# Herbage Mass, N Concentration, and N Uptake of Temperate Grasslands Can Adequately Be Estimated from UAV-Based Image Data Using Machine Learning

Ulrike Lussem [1,*], Andreas Bolten [1], Ireneusz Kleppert [1], Jörg Jasper [2], Martin Leon Gnyp [2], Jürgen Schellberg [3] and Georg Bareth [1]

1    Institute of Geography, University of Cologne, 50923 Cologne, Germany; andreas.bolten@uni-koeln.de (A.B.); iklepper@smail.uni-koeln.de (I.K.); g.bareth@uni-koeln.de (G.B.)
2    Research Center Hanninghof, Yara International ASA, 48249 Dülmen, Germany; joerg.jasper@yara.com (J.J.); martin.gnyp@yara.com (M.L.G.)
3    INRES—Institute of Crop Science and Resource Conservation, University of Bonn, 53113 Bonn, Germany; jk.schellberg@gmail.com
*    Correspondence: ulrike.lussem@uni-koeln.de

**Abstract:** Precise and timely information on biomass yield and nitrogen uptake in intensively managed grasslands are essential for sustainable management decisions. Imaging sensors mounted on unmanned aerial vehicles (UAVs) along with photogrammetric structure-from-motion processing can provide timely data on crop traits rapidly and non-destructively with a high spatial resolution. The aim of this multi-temporal field study is to estimate aboveground dry matter yield (DMY), nitrogen concentration (N%) and uptake (Nup) of temperate grasslands from UAV-based image data using machine learning (ML) algorithms. The study is based on a two-year dataset from an experimental grassland trial. The experimental setup regarding climate conditions, N fertilizer treatments and slope yielded substantial variations in the dataset, covering a considerable amount of naturally occurring differences in the biomass and N status of grasslands in temperate regions with similar management strategies. Linear regression models and three ML algorithms, namely, random forest (RF), support vector machine (SVM), and partial least squares (PLS) regression were compared with and without a combination of both structural (sward height; SH) and spectral (vegetation indices and single bands) features. Prediction accuracy was quantified using a 10-fold 5-repeat cross-validation (CV) procedure. The results show a significant improvement of prediction accuracy when all structural and spectral features are combined, regardless of the algorithm. The PLS models were outperformed by their respective RF and SVM counterparts. At best, DMY was predicted with a median $RMSE_{CV}$ of 197 kg ha$^{-1}$, N% with a median $RMSE_{CV}$ of 0.32%, and Nup with a median $RMSE_{CV}$ of 7 kg ha$^{-1}$. Furthermore, computationally less expensive models incorporating, e.g., only the single multispectral camera bands and SH metrics, or selected features based on variable importance achieved comparable results to the overall best models.

**Keywords:** grassland; multispectral; biomass; canopy height; N; concentration; uptake

## 1. Introduction

Grasslands are one of the major terrestrial biomes providing important ecological and economic services [1]. To name but a few, grasslands provide forage for ruminants for milk and meat production, are a source for biofuels and fibers, serve as a wildlife habitat, prevent soil erosion, play an important role in carbon sequestration, and, finally, also contribute to water purification [2]. Thus, grasslands substantially contribute to climate protection and food security [3–5]. In the European Union, grasslands account for 31.2% of the utilized agricultural area [6] and are mainly used for fodder and forage production [7].

To safeguard these ecosystem services, agricultural management needs to adjust to site-specific characteristics of grasslands to prevent, e.g., nutrient leaching originating from massive use of fertilizer, while simultaneously securing grassland production in adequate quantity and quality [2]. Both dimensions, expressed as yield (quantity) on the one hand and digestibility, protein content, fibers, or metabolizable energy (quality) on the other hand, are strongly related to plant N concentration. N is among the key parameters limiting agricultural production but also ecosystem functioning [8]. Precise information about herbage yield and plant N concentration is key to calculating N uptake and to specify required fertilizer rates, in order to prevent overuse of fertilizer and subsequent N leaching and eutrophication [9,10]. This is especially important in light of the EU Nitrates Directive, which obligates the member states to meet certain N application standards [11].

Thus, the sustainable and profitable management of grasslands relies on timely information on the quantity and quality of the sward to support management decisions, such as grazing rotation, timing of harvests, and fertilizer application [12,13]. Precision agriculture (PA) technologies are widely used to meet tasks regarding site-specific management, not only in arable crops [14,15], but also in grasslands [12].

However, PA technologies for herbage yield and quality assessment in high spatial and temporal resolution are still in the development phase due to the immanent spatio-temporal heterogeneity and variability of grasslands [13,16]. Different species and plant organs contribute to herbage yield throughout the growing season. Additionally, use intensity (cutting or stocking rates), fertilization, soil, climate, and weather characteristics have a strong impact on growth rates, yield, and N uptake [17–19]. The proper assessment of herbage yield and nutrient status traditionally implies destructive biomass sampling and subsequent laboratory analysis, such as wet chemistry or near-infrared spectroscopy (NIRS). However, laboratory analysis is costly, labor-intensive, and time consuming, and thus less suitable for timely and site-specific management decisions [20].

Non-destructive methods for herbage yield assessment, such as simple rulers or rising plate meters, which exploit the allometric relationship between (compressed) sward height and biomass, have already been developed [21–23]. Technologically, more advanced methods utilize LEDs or ultrasound to measure sward height [24,25]. Furthermore, a variety of vehicle-mounted or handheld proximal sensing applications exist to derive estimates of biomass and plant nutrient status exploiting canopy reflectance, such as the Yara N-Sensor [26,27], Trimble GreenSeeker [28], or general field spectroradiometers [29,30]. These methods have limitations, such as insufficiently representing spatial variability, allowing operator bias, requiring a high number of repetitions, and needing vehicles and direct access to the field, thereby potentially disturbing the canopy [31].

Alternatively, remote sensing technology has the potential to assess important plant traits (such as aboveground biomass, yield, and N status) with high spatial precision and on variable spatial scales [15,32–34]. Extensive overviews of remote sensing-based applications in grassland management are provided by Schellberg et al. [16], Wachendorf et al. [35], and Reinermann et al. [36]. Since the 1980s, mainly data from satellites and aircraft (active and passive sensors) have been employed for this purpose. Nevertheless, satellite-based data have limitations in terms of spatial resolution versus costliness, repetition rates, and weather conditions [37].

In the past 15 years, however, due to technological innovations in sensor and platform design, the use of unmanned aerial vehicles (UAV) and small high-resolution camera-systems in various spectral ranges (visible to near-infrared to shortwave infrared) has become more and more important in research and practical farming. Their user-friendliness and flexibility ensure data acquisition aligned to particular management needs to deliver information on plant growth and status with high spatial and temporal resolution [38,39]. Furthermore, the developments in structure-from-motion (SfM) and Multiview stereopsis (MVS) provide tools to easily derive spatially explicit 3D data from UAV-based image data [40–43]. These methods have been effectively applied to grassland monitoring by utilizing UAV-mounted imaging sensors. The majority of these studies focus on estimating

swarer height or dry matter yield by means of structural or spectral features, or a combination of both [44–54]. Only a few studies addressed bio-chemical parameters, such as N concentration, N uptake or fixation, crude protein, or acid detergent fiber (i.e., quality parameters) [49,55–60].

The combination of structural and spectral features improved the estimation accuracy for biomass and N concentration; Viljanen et al. [61] stated a slight improvement in DMY estimation when combining structural and spectral features using random forest and multiple linear regression. In the study by Pranga et al. [62], the combination of spectral and structural features from a multispectral camera using random forest provided the best results for the DMY estimation of perennial ryegrass with an RMSE of 382 kg ha$^{-1}$. Karunaratne et al. [63] reported a consistent improvement of DMY estimation when combining vegetation indices and 3D features over different flying heights. Oliveira et al. [59] reported an improvement for DMY estimation when using 3D features and spectral features from a hyperspectral camera, while spectral features from an RGB and multispectral camera in combination with 3D features provided the best results for the estimation of N concentration and uptake. The potential of combining features of different dimensions is therefore evident. To process the large quantity of data collected by UAV-based sensors, machine learning (ML) algorithms have gained popularity in recent decades. ML algorithms can handle multicollinearity (highly correlated features), high dimensional datasets, and non-linear relationships.

All in all, the potential of UAV-based sensors, the combination of structural and spectral features, and the use of ML algorithms to estimate the quality and quantity parameters of grasslands is evident. However, the above-mentioned studies are limited in growth stages (i.e., sample sizes) and the estimation of N concentration and uptake by UAV-based data is underrepresented in the current literature, since mainly ground-based approaches have been investigated to date. The present study aims to improve on these aspects.

Therefore, this study investigates UAV-based structural and spectral features from multispectral and RGB imaging sensors to quantify three important grassland management parameters: herbage dry matter yield (DMY), N concentration (N%), and N uptake (Nup), for an experimental grassland site in Germany based on data obtained over two years. The specific aims of this study are:

(i)     To develop estimation models for DMY, N%, and Nup utilizing three ML algorithms (namely, partial least squares, support vector machines, and random forest).
(ii)    To compare the prediction accuracy of the developed models with and without structural features.
(iii)   To identify potential key variables (most important features) for the estimation models.

## 2. Materials and Methods

### 2.1. Study Site

This study is based on data acquired from a grassland field trial in Germany. The site is located in Neunkirchen-Seelscheid (Bergisches Land region, North Rhine-Westphalia), about 30 km southeast of Cologne (50.858986N, 7.312736E). The mean annual temperature is 10 °C and mean annual precipitation 800 mm. The mean temperature in 2018 and 2019 was significantly higher than the long-term average (11.7 and 11.4 °C, respectively), and annual precipitation was lower (628 and 782 mm, respectively). As can be seen in Figure 1, distinct arid periods occur in the summer months.

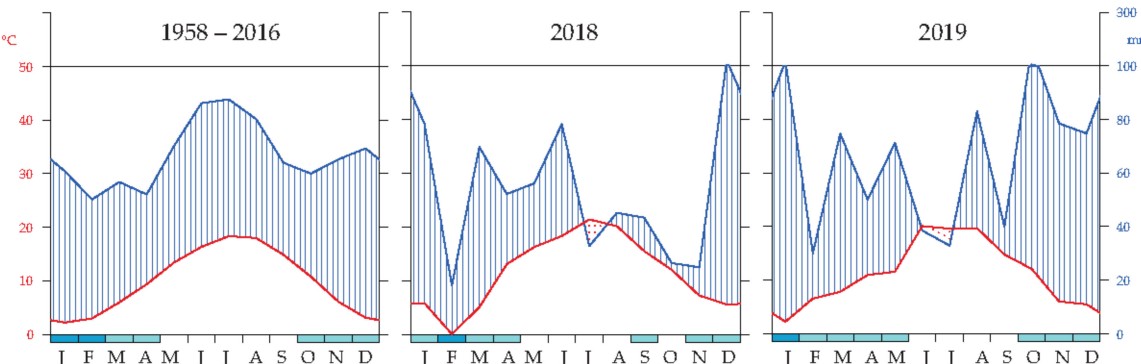

**Figure 1.** Walter–Lieth climate diagram from the German Weather Service (DWD) weather station (Cologne Bonn airport, station ID 2667) for the long-term average (1958–2016) and the sampling years 2018 and 2019.

The field experiment was established in March 2017 on a conventionally managed grassland field with a significant slope. The vegetation type was identified as Lolio-Cynosuretum and the main species were *Lolium perenne*, *Alopecurus* sp., and *Festuca* sp., among others. The field trial comprised three N-treatment levels (50, 100, 150 kg N ha$^{-1}$ for growing periods one and two, and 25, 50, and 75 kg N ha$^{-1}$ for growing period three) and one control treatment with no fertilizer application. Each treatment had 39 replicates arranged in a chessboard-like pattern, resulting in 156 plots. Each plot covered an area of 6 × 6 m. Figure 2 shows the layout of the experimental field.

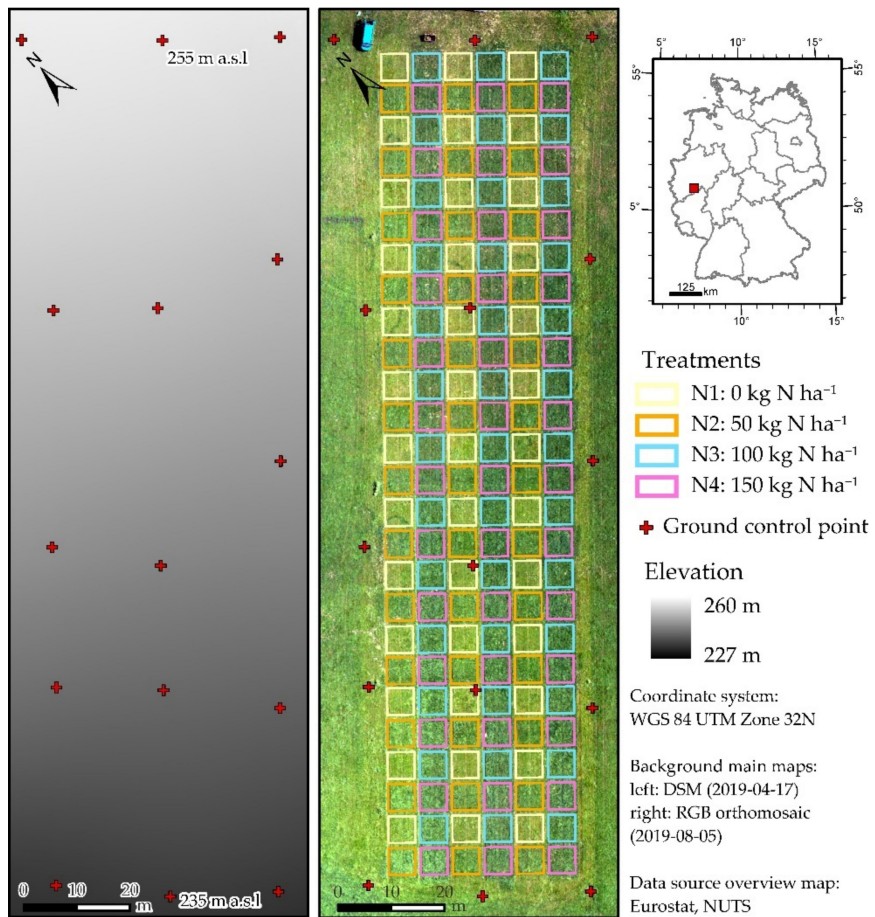

**Figure 2.** Map of study site Neunkirchen-Seelscheid (50.858986N, 7.312736E), Germany. (**Left**) Digital surface model. (**Right**) RGB orthomosaic.

The site was set up as a three-cut system. The first cut was harvested in May, the second in July/August, and the third in October, similar to local farming practice. To obtain field reference data, a conventional lawn mower (Viking Model MB_756_YC, STIHL AG & Co. KG, Dieburg, Germany) was used to harvest a 0.54 × 5.5 m wide strip of standing biomass from each of the 156 plots (see Figure 3c). The fresh biomass samples were weighed directly in the field and a subsample of about 300 g was obtained. The subsamples were dried in a forced-air dryer to a constant weight for three days at 65 °C. Subsequently, the weights of the dried subsamples were used to calculate the dry matter yield (DMY) in kg per hectare. N% was determined by the Kjeldahl method, extracting complete N. Nup was calculated from DMY and N% values for every plot ((DMY × N%)/100).

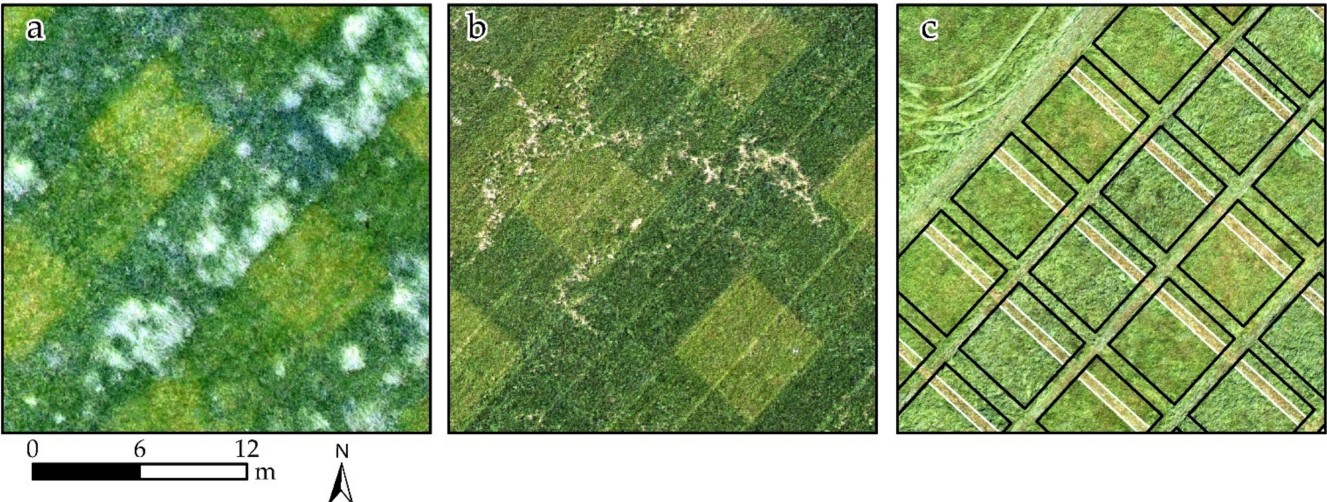

**Figure 3.** Close-up views of (**a**) bending and lodging canopy (22 May 2019), (**b**) rodent activity (23 July 2019), and (**c**) sampling area (white rectangles) within quadratic plots (3 July 2018).

## 2.2. Sensors and Platform

A multirotor UAV, (MK Okto XL 6S12, HiSystems GmbH, Moormerland, Germany) was used as a carrier platform for two cameras. RGB (visible spectral range) image data were recorded with a Sony Alpha 7r with 36 megapixels (MP) and a Zeiss Batis 25 mm lens. Multispectral image data were recorded with a Micasense RedEdge-M camera (MS) (AgEagle Sensor Systems Inc., d/b/a MicaSense, Wichita, KS, USA). The MS camera was equipped with five different sensors (1.2 MP each) in the visible-to-near-infrared (VNIR) region of the electromagnetic spectrum (see Figure 4 and Table 1). Additionally, the camera had an irradiance sensor and a GPS module.

**Table 1.** Band specifications of the MicaSense RedEdge-M camera. RedEdge-M user manual [64].

| Band Number | Band Name | Center Wavelength (nm) | Bandwidth FWHM (nm) |
|:---:|:---:|:---:|:---:|
| 1 | Blue | 475 | 20 |
| 2 | Green | 560 | 20 |
| 3 | Red | 668 | 10 |
| 4 | Near IR | 840 | 40 |
| 5 | Red Edge | 717 | 10 |

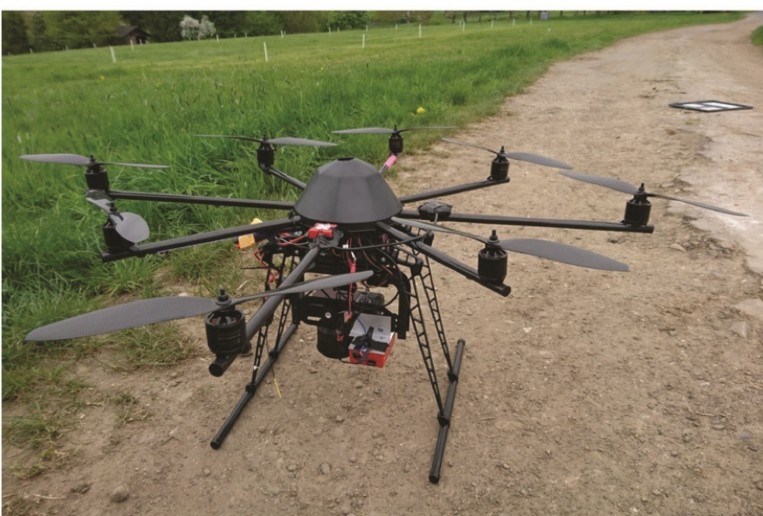
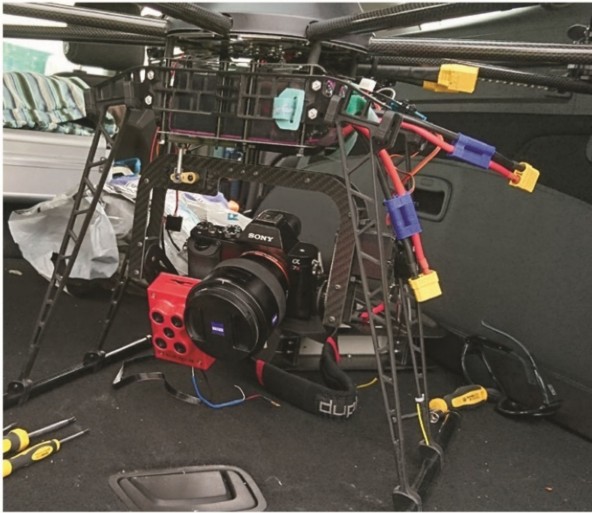

**Figure 4.** Multirotor UAV equipped with MicaSense RedEdge-M and Sony Alpha 7r cameras.

### 2.3. UAV-Based Data Acquisition

Flight campaigns for biomass sampling dates ($T_S$) were scheduled either one day in advance of the harvest or on the same day between 10 and 11 a.m. (before solar noon). To obtain the base models for sward height calculations for each growing period, flight campaigns with the RGB camera were scheduled after the complete harvest ($T_0$) of the field. The image overlap for the MS camera was approximately 75% forward and 70% sideward (FOV diagonal 57°). The Sony camera had approximately 80% overlap in both directions (FOV diagonal 84°). Flight height of the UAV was set to 35 m above ground, following the terrain, resulting in a ground sampling distance (GSD) of ~0.7 cm for the RGB and ~2.3 cm for the MS camera. For accurate georeferencing, 15 ground control points (GCPs) were evenly distributed across the site (see Figure 2). The GCPs were measured on each sampling date with a real-time kinematic differential GPS (GR-5, Topcon, Japan). Before and after each flight, the radiometric calibration panel (gray standard provided by the manufacturer) was recorded with the MS camera to provide reflectance values for subsequent radiometric calibration.

For an additional radiometric assessment of the MS camera, six panels (80 × 80 cm) in different shades of gray were laid out at the experimental site (see Figure 5). The panels were coated with a matte color, exhibiting near-Lambertian reflectance properties (NEXTEL® color with a proportion of 100, 75, and 50, and 25, 10, and 0% black). Unfortunately, the panels were not available for all sampling dates and the 10 and 0% black panels exhibited extreme saturation. The reflectance of each panel was measured with an ASD FieldSpec 3 spectroradiometer (Malvern Panalytical Ltd., Cambridge, UK), which records reflectance in the range of 350–2500 nm. Subsequently, spectroradiometer readings were summarized to fit the bands of the MS camera. The data from the NEXTEL-coated panels were not included into the radiometric calibration workflow of the MS data.

To summarize the field sampling campaigns, Figure 6 shows a schematic overview of the sampling dates for UAV-based data acquisition and for destructive biomass sampling.

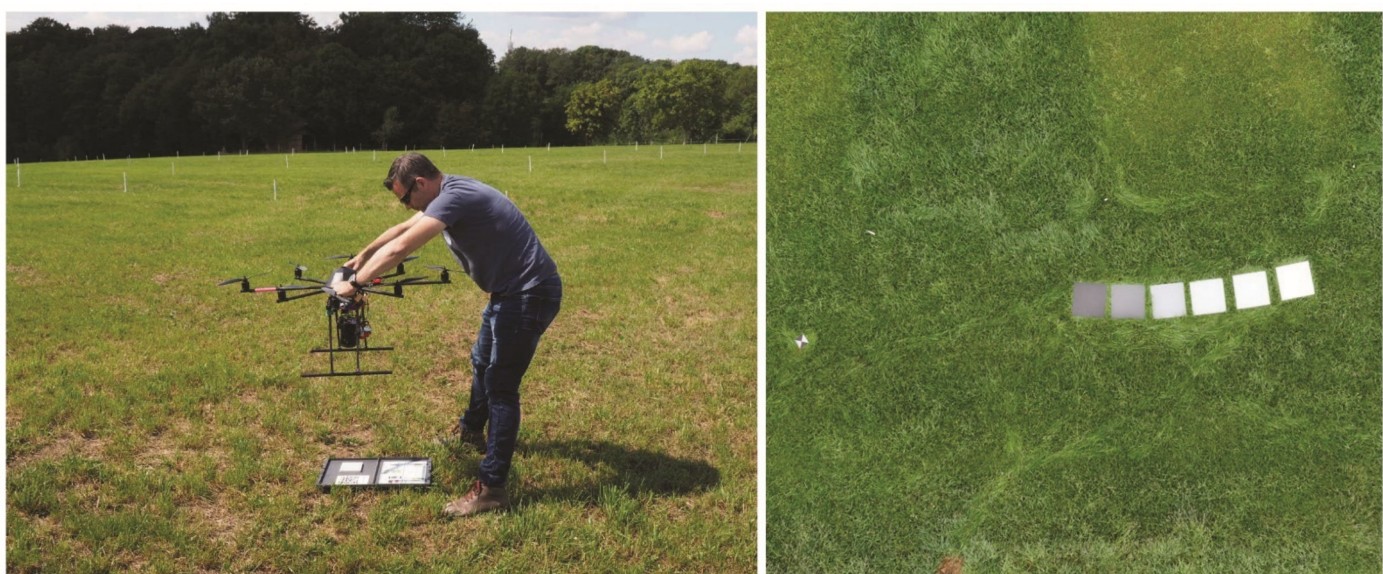

**Figure 5.** Acquiring images of the MicaSense RedEdge-M radiometric calibration panel ((**left**); image credits: Christoph Hütt), and additional gray panels for radiometric assessment and ground control point (**right**).

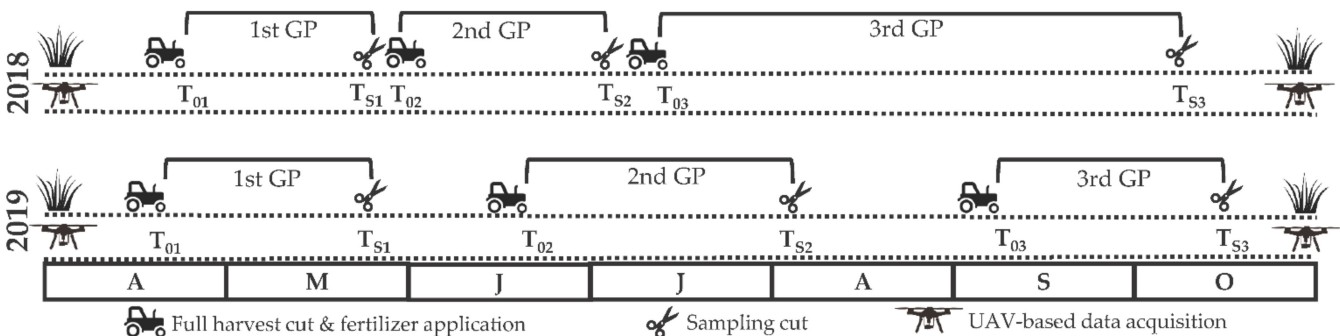

**Figure 6.** Schematic overview of sampling dates for UAV-based data acquisition and destructive biomass sampling. $T_0$: base model after equalization cut, $T_s$: sampling date, GP: growing period.

### 2.4. UAV-Based Data Processing and Feature Extraction

The relevant steps of data processing, feature extraction, and model building and assessment are depicted in the schematic workflow in Figure 7.

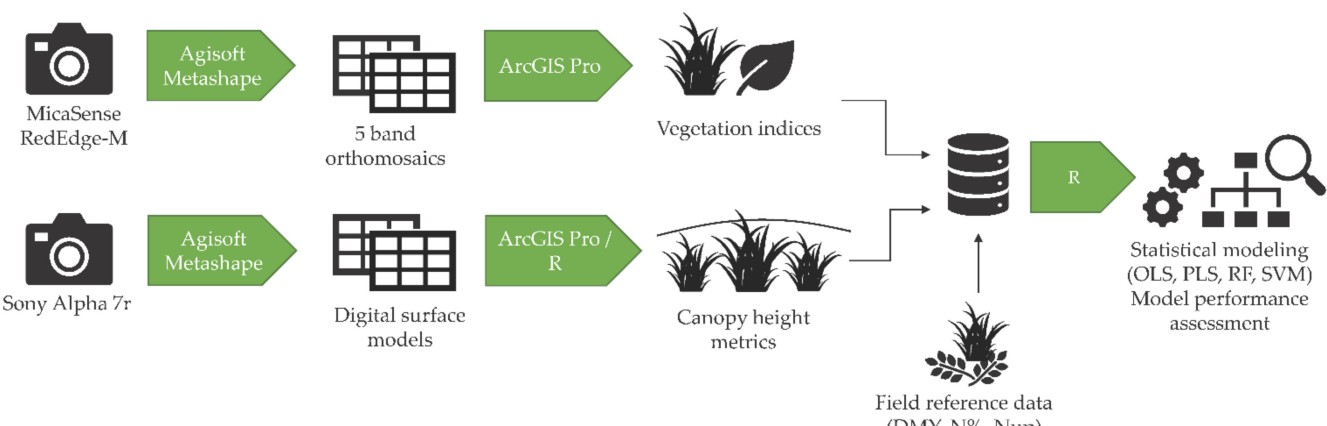

**Figure 7.** Schematic workflow of data acquisition, processing, feature extraction, and modeling.

Image data were processed in the SfM software Agisoft Metashape (Agisoft LLC, St. Petersburg, Russia). After loading the images, at least ten markers were placed for each GCP. After image alignment (*high*-quality setting) the dense cloud was computed using the *high*-quality setting with *aggressive* depth filtering for the base models and *mild* depth filtering for the sampling dates to preserve the finer details of the sward. The datasets of the MS camera were then radiometrically calibrated by the *calibrate reflectance* function using the calibration factors of the irradiance sensor and the gray reference panel. After computing the dense point cloud, the DSM and orthomosaic were compiled. The DSMs of the RGB camera were exported in a resolution of 1 cm and the orthomosaics of the MS camera were exported with a 2.5 cm resolution.

To obtain the estimates of UAV-based sward height, the base model DSM of each growing period was subtracted from the sampling date DSMs of the respective growing period using ArcGIS Pro (ESRI, Redlands, CA, USA). In a second step, the following sward height (SH) metrics were calculated in the statistical computation software R [65] using R Studio GUI [66]: mean, minimum, maximum, 90th, 75th, 50th (median), and 25th percentiles (SHmean, SHmin, SHmax, SHp90, SHp75, SHp50, SHp25). The graphical modeling interface *model builder* in ArcGIS Pro was used for vegetation index (VI) calculations from the raster bands of the MS orthomosaics. A total of 19 indices were selected based on their characterization of biochemical and structural traits of vegetation in order to be comparable to existing studies. For each date, a polygonal shapefile was created for the biomass sampling area per plot based on orthomosaics obtained directly after biomass sampling (see Figure 8). The respective shapefiles were used to extract height and spectral features by zonal statistics for each plot in ArcGIS Pro for further analysis in R. Table 2 lists the 15 VIs based on the visible and near-infrared region, hereafter referred to as VI.ms.

**Table 2.** Vegetation indices derived from the visible-to-near-infrared spectral region.

| Name | Equation | Application | Reference |
|------|----------|-------------|-----------|
| Normalized Difference Vegetation Index | $\text{NDVI} = \frac{(\text{NIR}-\text{R})}{(\text{NIR}+\text{R})}$ | Greenness, green biomass, phenology | [67] |
| Green Normalized Difference Vegetation Index | $\text{GNDVI} = \frac{(\text{NIR}-\text{G})}{(\text{NIR}+\text{G})}$ | Green biomass, N concentration, LAI | [68] |
| Blue Normalized Difference Vegetation Index | $\text{BNDVI} = \frac{(\text{NIR}-\text{B})}{(\text{NIR}+\text{B})}$ | Greenness, green biomass, phenology | [69] |
| Optimized Soil-Adjusted Vegetation Index | $\text{OSAVI} = \frac{1+0.16(\text{NIR}-\text{R})}{\text{NIR}+\text{R}+0.16}$ | Green biomass, photosynthesis rate | [70] |
| Modified Soil-Adjusted Vegetation Index | $\text{MSAVI} = 0.5\left[2*\text{NIR}+1-\sqrt{(2\text{NIR}+1)^2-8(\text{NIR}-\text{R})}\right]$ | Green biomass, photosynthesis rate | [71] |
| Modified Chlorophyll Absorption in Reflectance Index 1 | $\text{MCARI1} = 1.2[2.5(\text{NIR}-\text{R})-1.3(\text{NIR}-\text{G})]$ | Chlorophyll concentration, plant stress, photosynthesis rate | [72] |
| Enhanced Vegetation Index | $\text{EVI} = 2.5\frac{(\text{NIR}-\text{R})}{\text{NIR}+6\text{R}-7.5\text{B}+1}$ | Green biomass, greenness, phenology | [73] |
| Normalized Difference Red Edge Index | $\text{NDREI} = \frac{(\text{NIR}-\text{RE})}{(\text{NIR}+\text{RE})}$ | Chlorophyll | [74] |

**Table 2.** *Cont.*

| Name | Equation | Application | Reference |
|---|---|---|---|
| Renormalized Difference Vegetation Index | $RDVI = \frac{(NIR-RE)}{\sqrt{(NIR+RE)}}$ | Green biomass | [75] |
| Simple Ratio | $SR = \frac{NIR}{R}$ | Green biomass | [76] |
| (Simplified) Canopy Chlorophyll Content Index | $CCCI = \frac{NIR-RE}{NIR+RE} / \frac{NIR-R}{NIR+R}$ | Chlorophyll concentration, photosynthesis | [77] modified by [78] |
| Modified Triangular Vegetation Index 2 | $MTVI2 = \frac{1.5[1.2(NIR-G)-2.5(R-G)]}{\sqrt{(2NIR+1)^2-(6NIR-5\sqrt{R})-0.5}}$ | Green biomass | [72] |
| Modified Simple Ratio | $MSR = \frac{NIR}{R} + 1/\sqrt{\frac{NIR}{R} + 1}$ | Green biomass | [79] |
| Near-Infrared to Red Edge Ratio | $NIR.RE = \frac{NIR}{RE}$ | Chlorophyll, N | [80] |
| Red Edge to Red Ratio | $RE.R = \frac{RE}{R}$ | Chlorophyll, N | [80] |

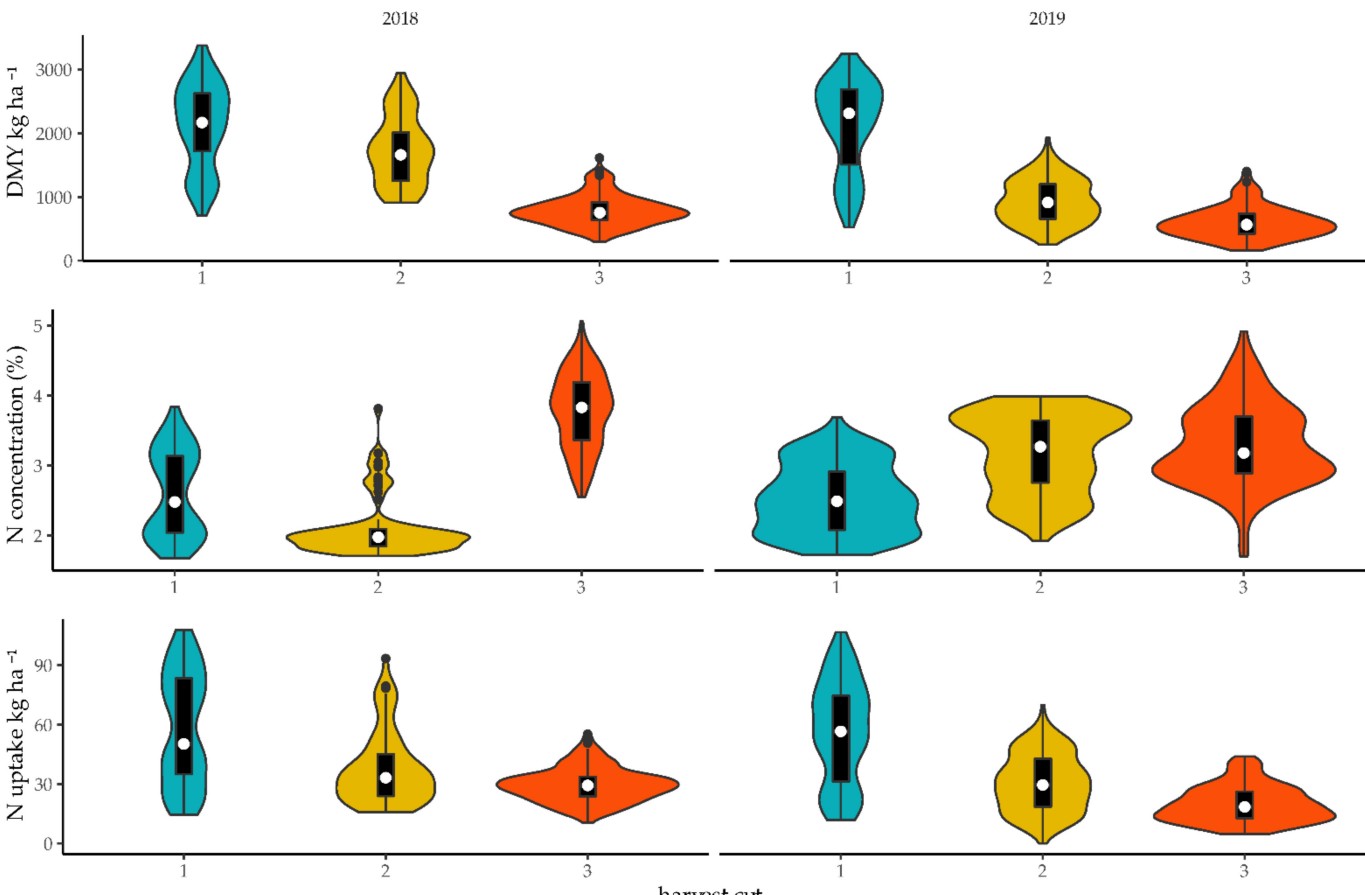

**Figure 8.** Violin plots of DMY, N%, and Nup by harvest cut and year. Violins indicate point density, black boxes show the 25. and 75. percentiles, whiskers show the 5 and 95 percentiles, and white circles represent the median.

Additional to the VI.ms, four indices from the visible spectrum were selected, hereafter referred to as VI.rgb (see Table 3). We decided to calculate the VI.rgb from the bands of the MS camera since no radiometric correction was applied to the RGB camera data.

Comparability between sampling dates with varying irradiation is rarely possible with an uncalibrated RGB camera, as shown by Lussem et al. 2019.

**Table 3.** Vegetation indices derived from the visible spectral region.

| Name | Equation | Application | Reference |
|------|----------|-------------|-----------|
| Normalized Green Red Difference Index | $NGRDI = \frac{(G-R)}{(G+R)}$ | Green biomass | [81] |
| Plant Pigment Ratio Index | $PPRI = \frac{(G-B)}{(G+B)}$ | Chlorophyll | [82] |
| Red Green Blue Vegetation Index | $RGBVI = \frac{(G^2-(R*B))}{(G^2+(R*B))}$ | Green biomass | [30] |
| Visible Atmospherically Resistant Index | $VARI = \frac{(G-R)}{(G+R-B)}$ | Green biomass | [83] |

### 2.5. Statistical Analysis

UAV-based structural (height) and spectral (reflectance) information of the canopy and their combination were set as independent variables to estimate DMY, N%, and Nup based on the pooled dataset from 2018 and 2019. To test the predictive power of the extracted features, empirical models using a linear regression model (LM) and three machine learning algorithms were compared, i.e., partial least squares regression (PLS), random forest regression (RF), and support vector machine regression (SVM). The non-parametric algorithms were chosen to (i) handle multicollinearity and non-linear relationships present in the data, and (ii) to be comparable to similar studies.

Table 4 presents the feature sets considered in the analysis. Statistical analysis was performed in R. The package *caret* was chosen as a modeling framework, since it provides cross-validation procedures and most ML algorithms can be implemented [84,85].

**Table 4.** Feature sets.

| Name | Description | Features Included |
|------|-------------|-------------------|
| SH | Sward height metrics | SHmean, SHmin, SHmax, SHp90, SHp75, SHp50, SHp25 |
| VI.ms | Vegetation indices visible to near-infrared spectrum | See Table 2 |
| SB | Single bands of the Micasense RedEdge-M | Blue, green, red, red edge, near-infrared (B, G, R, RE, NIR) |
| VI.rgb | Vegetation indices visible spectrum | See Table 3 |

PLS is a generalization of multivariate linear regression, but with the ability to handle the collinearity of multiple predictor variables. It also linearly combines and thus reduces the predictor variables to a few latent vectors, which cover the maximum of covariance between response and predictor variables [86]. Random forest (RF) is an ensemble algorithm [87] and has become popular in remote sensing research because of its good predictive performance [88]. For regression, a large number of decision trees was constructed with a random selection of variable subsets and the mean prediction of all trees was taken as the output. The support vector machine (SVM) algorithm was developed by [89]. SVMs have gained popularity in remote sensing due to their capacity to generalize well, even with small sample sizes [90]. SVMs map the covariates into a high-dimensional feature space using a kernel trick to determine an optimal separating hyperplane between data points. Both linear and non-linear kernels can be implemented, depending on the relationship of the data [91].

Hyperparameter Tuning and Model Assessment

To achieve reliable estimates, the algorithm-specific hyperparameters should be tuned (i.e., optimized) [92,93]. Thus, for each algorithm and feature set combination, some of the respective hyperparameters were determined using search parameters within a cross-validation process. Final parameters were chosen based on the lowest RMSE and

implemented in the final model. We used the *pls* package [94] for PLS modeling. The algorithm-specific hyperparameter of PLS is the number of components (*ncomp*) to be considered. *Ncomp* was tuned using all possible numbers for each feature set combination (e.g., for a model with 31 features, *ncomp* was set from 2 to 31).

RF hyperparameters include *mtry*, the number of randomly selected variables at each split, and *ntree*, the number of decision trees built. To find the best value for *mtry*, a search was performed over all possible numbers for each feature set combination (e.g., for a model with 31 features, the search for mtry was set from 2 to 31). *Ntree* was set to 500, as commonly recommended [88]. Tests with a higher number of trees did not yield significantly better results. The minimum number of observations per node (minimum node size) was set to five and the splitrule to *extratrees (extremely randomized trees)*. The importance measure was set to permutation, as recommended for RF regression analysis [95]. The *ranger* package [96] was chosen as the RF algorithm. *Ranger* is a computationally faster implementation of the original random forest algorithm by Breiman [87].

For the SVM algorithm, a radial basis function kernel was chosen, implemented in the *kernlab* package [97], to account for non-linear relationships present in the data. A search over the cost parameter C was performed with C equal to 0.25, 0.5, 1, 2, or 4. Parameter C allows a trade-off between training error and model complexity, and thus reduces overfitting [91].

For linear models, all features were analyzed separately; whereas, for the ML algorithms, the feature sets SH, SB, VI.rgb, and VI.ms were either analyzed separately or in combination (SH_SB, SH_VI.rgb, SH_VI.ms). Finally, all feature sets were combined (SH_SB_VI.rgb_VI.ms) for analysis. Since no genuinely independent test dataset was available (i.e., from a different site), models were built using 10-fold cross-validation with 5 repeats, resulting in 50 model runs per algorithm and feature set. The prediction accuracy was quantified by calculating the cross-validation (CV) coefficient of determination ($R^2_{CV}$) and root-mean-square error ($RMSE_{CV}$) for each model run to assess the models in terms of precision and variability. Additionally, the relative (or normalized) RMSE was calculated using the mean value of the respective response variable as the divisor.

Furthermore, variable importance was calculated for all models to determine the important features of the feature set combinations and evaluate the stability of feature importance over different models. To achieve comparability between the different ML algorithms, the variable importance was calculated by permutation, which is a model agnostic method and can be applied to any kind of model. This approach was implemented in the R package *vip* [98,99]. The permutation-based variable importance was defined as the decrease in a chosen error metric (here RMSE) when a single variable was randomly shuffled to break the connection to the response variable. Iteratively, all features were permuted and the permuted result was compared to the original result. Thus, the more the RMSE increased, the more important a specific feature was for the model. This procedure was applied to all variables of all models and repeated 25 times per model to account for the stability of the outcome.

## 3. Results

For the statistical analysis, the dataset was filtered for plots with a lodging canopy. In 2018, 3 plots of the first growth, 84 plots of the second growth, and 1 plot of the third growth were excluded. In 2019, 16 plots of the first growth were excluded. Additionally, one observation with missing laboratory results was excluded. Overall, 832 observations were analyzed. In Figure 8, the selected images highlight the variability of the sward within one year (a and b) and the sampled areas within the quadratic plots (c).

### 3.1. Distribution of Response Variables and Correlation Analysis

The experimental setup of four treatments generated a wide range of biomass values (see Figure 8). However, due to severe drought in 2019, the biomass values were at a lower level for

the second and third cuts compared to 2018. The range of DMY was 300–3376 kg ha$^{-1}$ in 2018 and 158–3244 kg ha$^{-1}$ in 2019.

Figure 8 shows that the first cut has the highest variability in DMY for both years. Lowest variability in DMY was found in the third cut in both years (2018: 300–3376 kg ha$^{-1}$, 2019: 158–3244 kg ha$^{-1}$). The N concentration ranged from 1.67–5.06% in 2018, and 1.7–4.92% in 2019. N uptake ranged from 10–108 kg N ha$^{-1}$ in 2018, and 5–107 kg N ha$^{-1}$ in 2019 (see also Table 5).

The results of the bivariate linear correlation analysis of the pooled dataset are presented in a correlation map in Figure 9. DMY displays strong correlations with most canopy height metrics, moderate-to-strong correlations with most VI.ms, and weak correlations with VI.rgb and single bands, except for the NIR band. N% only shows very weak correlations to the predictor variables, with the highest, although negative, correlation to the PPRI. Nup displays a similar pattern in correlation strength as DMY, although slightly weaker. Furthermore, the predictor variables show a high degree of multicollinearity.

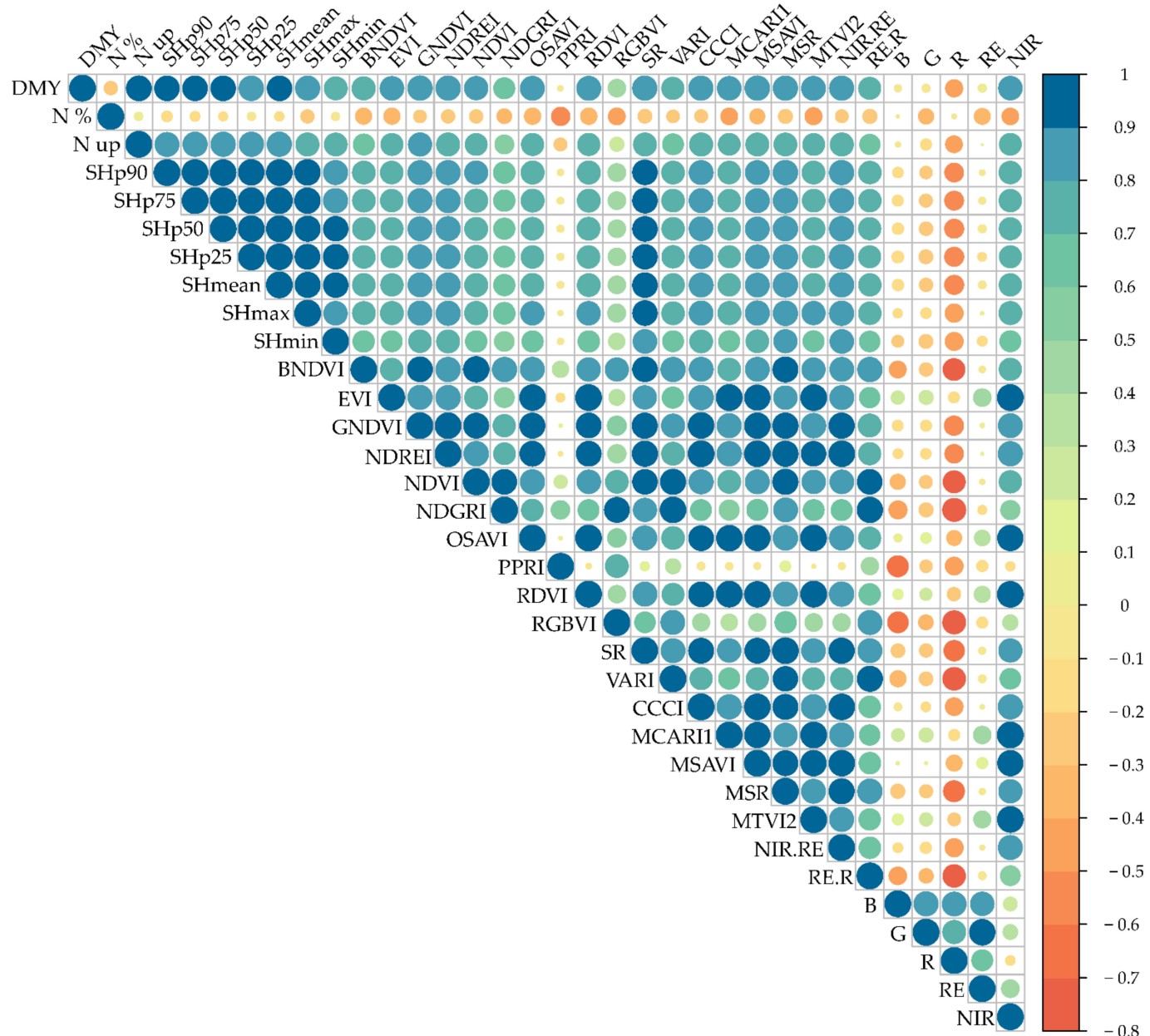

**Figure 9.** Correlation matrix (Pearson correlation coefficient) of response variables and prediction features (see Tables 2–4 for explanation of abbreviations).

**Table 5.** Descriptive statistics for the response variables DMY, N%, and Nup per year (max: maximum; min: minimum; sd: standard deviation).

| | DMY kg ha$^{-1}$ | | | | N% | | | | Nup kg N ha$^{-1}$ | | | |
|---|---|---|---|---|---|---|---|---|---|---|---|---|
| Year | Mean | Max | Min | sd | Mean | Max | Min | sd | Mean | Max | Min | sd |
| 2018 | 1503 | 3376 | 300 | 791 | 2.99 | 5.06 | 1.67 | 0.88 | 43 | 108 | 10 | 24 |
| 2019 | 1189 | 3244 | 158 | 796 | 3.01 | 4.92 | 1.70 | 0.64 | 35 | 107 | 5 | 23 |

### 3.2. Error Assessment of SfM/MVS Processing

Although all image datasets were processed using the same parameters, small differences in the processing results are apparent (see Table 6). The errors in the X- and Y-coordinates were 0.97–6.74 cm, while the height error ranged from 0.50–1.43 cm for the Sony $\alpha$ 7r datasets. The MS orthomosaics were calculated only for the sampling dates. In general, the MS data errors were less than the Sony data, especially the Z error.

**Table 6.** Error report of the photogrammetric data processing in Metashape for each UAV campaign: Sony Alpha 7r for canopy height feature's calculation; MicaSense Rededge-M for spectral features (only for sampling dates). $T_0$: date of base model acquisition; $T_S$: sampling date; GP: growing period.

| Year | GP | Date | Error (cm) Sony $\alpha$ 7r | | | Error (cm) MS RedEdge-M | | |
|---|---|---|---|---|---|---|---|---|
| | | | X | Y | Z | X | Y | Z |
| **2018** | 1 | $T_0$ (April 24) | 1.50 | 1.26 | 0.72 | - | - | - |
| | | $T_S$ (May 25) | 0.97 | 1.43 | 1.02 | 0.89 | 1.38 | 0.42 |
| | 2 | $T_0$ (May 29) | 2.72 | 1.83 | 0.62 | - | - | - |
| | | $T_S$ (May 02) | 1.96 | 1.57 | 0.50 | 0.80 | 1.28 | 0.35 |
| | 3 | $T_0$ (July 12) | 3.97 | 1.87 | 0.98 | - | - | - |
| | | $T_S$ (October 10) | 1.50 | 1.54 | 0.57 | 0.99 | 1.26 | 0.48 |
| **2019** | 1 | $T_0$ (April 17) | 1.65 | 1.60 | 0.93 | - | - | - |
| | | $T_S$ (May 22) | 1.60 | 1.66 | 0.99 | 1.02 | 1.00 | 0.22 |
| | 2 | $T_0$ (June 18) | 2.88 | 1.47 | 0.58 | - | - | - |
| | | $T_S$ (August 05) | 1.25 | 1.44 | 0.68 | 1.10 | 1.22 | 0.34 |
| | 3 | $T_0$ (September 02) | 6.74 | 3.93 | 1.43 | - | - | - |
| | | $T_S$ (October 14) | 3.87 | 2.43 | 0.77 | 0.94 | 1.39 | 0.23 |

### 3.3. Radiometric Assessment MS Camera

The radiometric assessment of the MS camera compared to the reflectance values of the reference panels measured by an ASD FieldSpec 3 spectroradiometer is presented in Figure 10 and Table 7. Unfortunately, the reference panels were not available for all sampling dates. In particular, the blue and red bands exhibit the lowest $R^2$ values with a distinct saturation effect above reflectance of 0.2. However, the reflectance of plants in the blue and red regions seldom exceeds this value; thus, it can be presumed that the MS camera provides reliable results. The NIR and RE bands have the closest relationship to the benchmark instrument, although the MS camera overestimates the NIR reflectance. However, on the third sampling date in 2019, the NIR channel was underestimated by the MS camera.

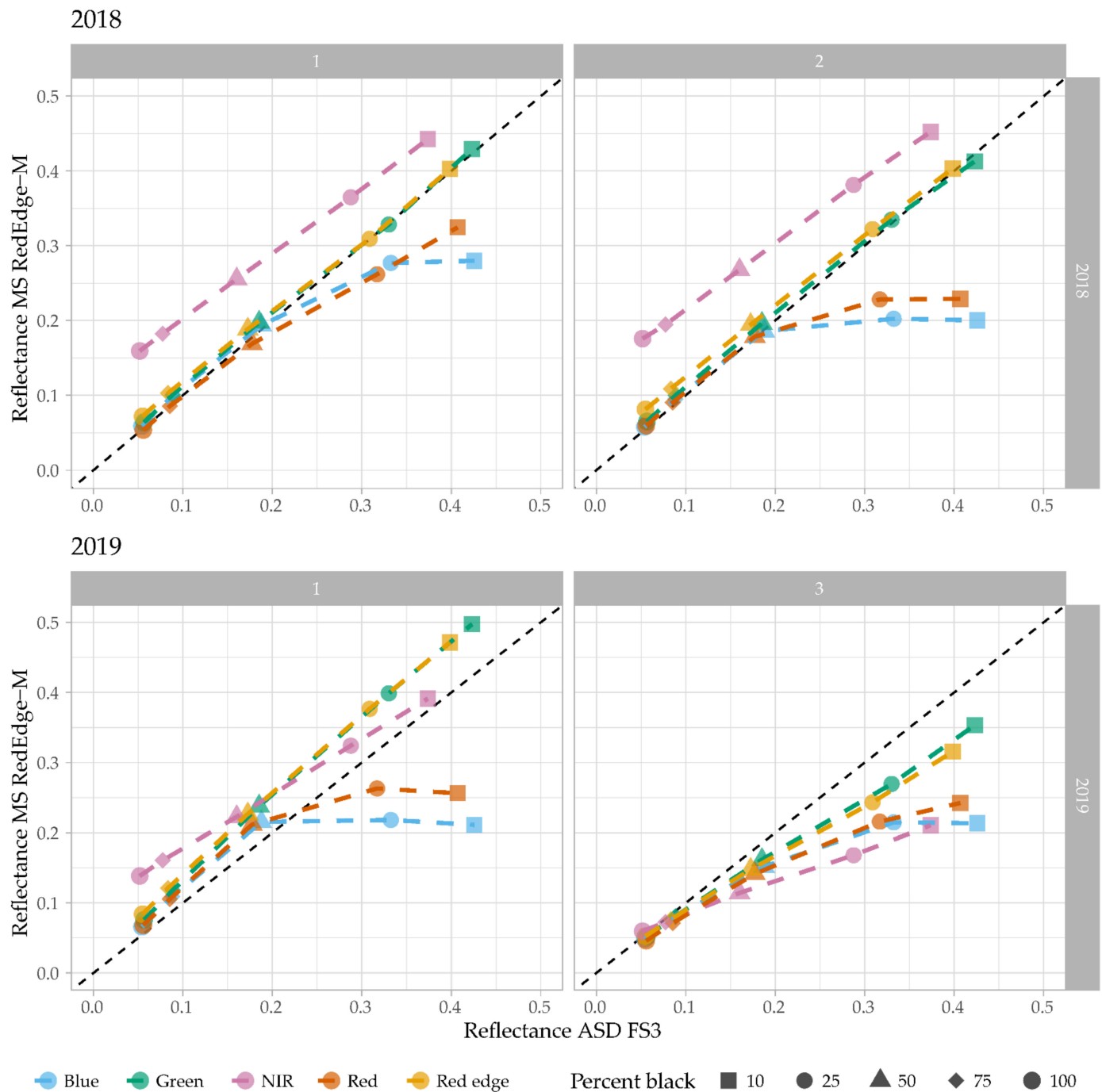

**Figure 10.** Scatterplot of values for MS camera and ASD Fieldspec 3 measurements of reference panels.

Table 7 presents the $R^2$ and RMSE between the spectroradiometer measurements and the MS camera of the reference panels. In particular, the blue and red bands show weaker relationships with the ASD FieldSpec3 measurements ($R^2$ 0.58–0.74 and 0.69–0.88, respectively). As mentioned above, these low $R^2$ values are due to the saturation effect above 0.2% reflectance. On the other hand, the red edge and NIR bands show the best agreement with the benchmarking instrument.

**Table 7.** Coefficient of determination $R^2$ and RMSE (% reflectance) of the MS RedEdge-M camera and ASD Fieldspec 3 measurements of reference panels for four sampling dates.

| | 2018 | | | | 2019 | | | |
| | May 25 | | July 2 | | May 22 | | October 14 | |
| | $R^2$ | RMSE | $R^2$ | RMSE | $R^2$ | RMSE | $R^2$ | RMSE |
|---|---|---|---|---|---|---|---|---|
| **Blue** | 0.74 | 0.17 | 0.58 | 0.22 | 0.71 | 0.20 | 0.71 | 0.21 |
| **Green** | 0.89 | 0.09 | 0.92 | 0.09 | 0.90 | 0.07 | 0.94 | 0.12 |
| **Red** | 0.88 | 0.13 | 0.69 | 0.19 | 0.83 | 0.15 | 0.80 | 0.19 |
| **Red Edge** | 0.93 | 0.08 | 1.00 | 0.02 | 0.95 | 0.06 | 0.96 | 0.11 |
| **NIR** | 1.00 | 0.09 | 1.00 | 0.10 | 1.00 | 0.06 | 1.00 | 0.15 |

### 3.4. Dry Matter Yield Prediction

In Figure 11, the results of all cross-validation model runs of the simple linear regression (SLR) are presented. The metrics SHmean, SHp90, SHp75, and SHp50 proved to be better predictors for the DMY estimation than all spectral features in simple linear regression based on the median $RMSE_{CV}$ (below 350 kg ha$^{-1}$). VI.ms (GNDVI, NDREI, SR, CCCI, MSAVI, MSR, and NIR.RE) all had a median $R^2_{CV}$ above 0.75 and $RMSE_{CV}$ below 400 kg ha$^{-1}$. The VI.rgb and single bands, except the NIR band, all exhibited very low $R^2_{CV}$ (almost zero) and high $RMSE_{CV}$.

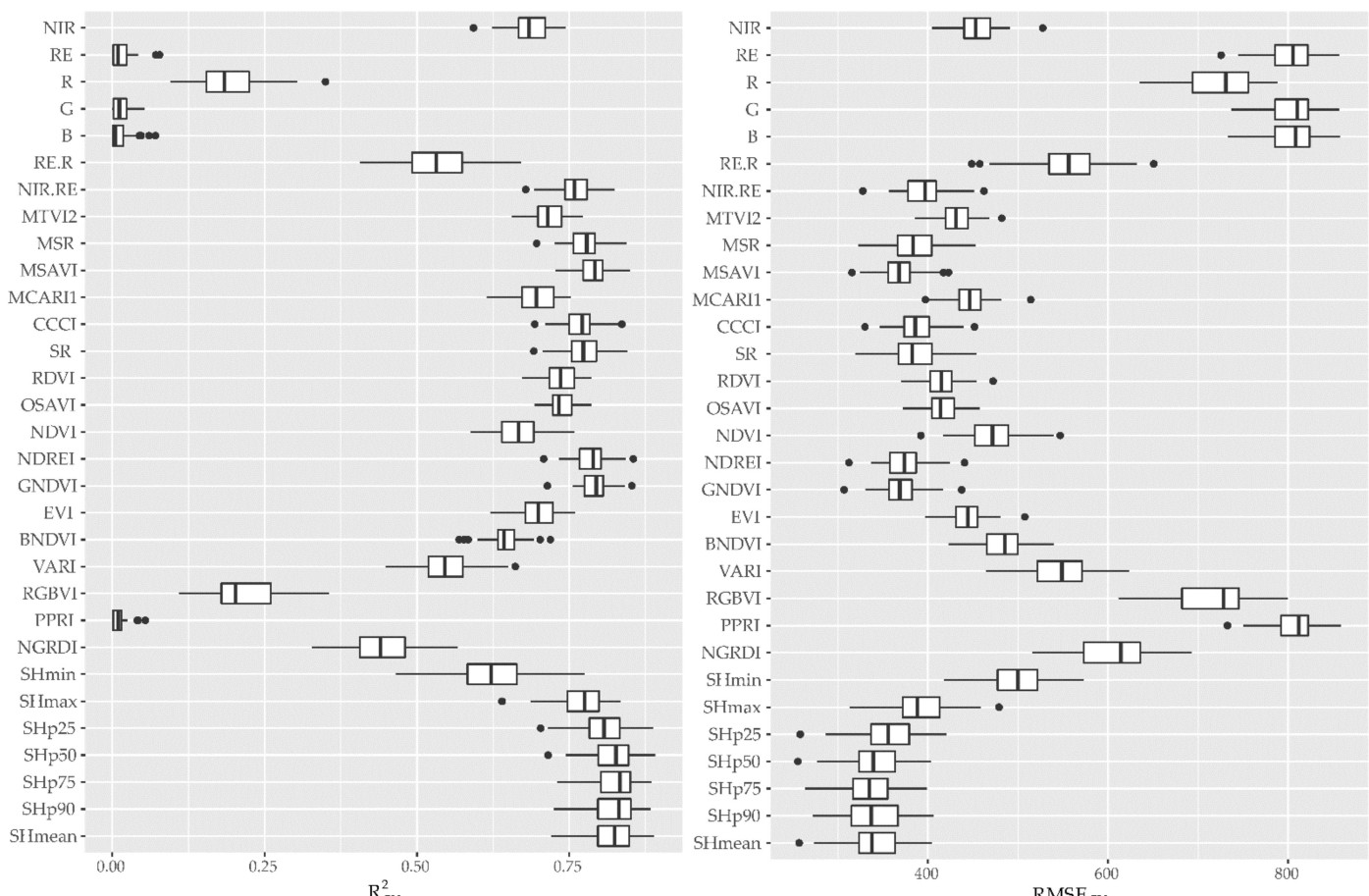

**Figure 11.** Box and whisker plot of $R^2_{CV}$ and $RMSE_{CV}$ (kg ha$^{-1}$) of all resamples of linear models (simple linear regression) for herbage yield prediction.

The results for the performance metrics of all cross-validation model runs of the three ML algorithms (PLS, SVM, RF) for the DMY estimation are shown in Figure 12, based on

different feature sets. In general, the PLS algorithm provides poorer results compared to the RF and SVM algorithms. Furthermore, the RGB feature set performed worse, irrespective of the modeling algorithm. The best models per algorithm were the combinations of all features and the SH_VI.ms feature combination. Although other feature combinations showed similar $R^2_{CV}$ values, the median RMSE$_{CV}$ was lower and less spread in these two mentioned combinations when comparing the performance by algorithm.

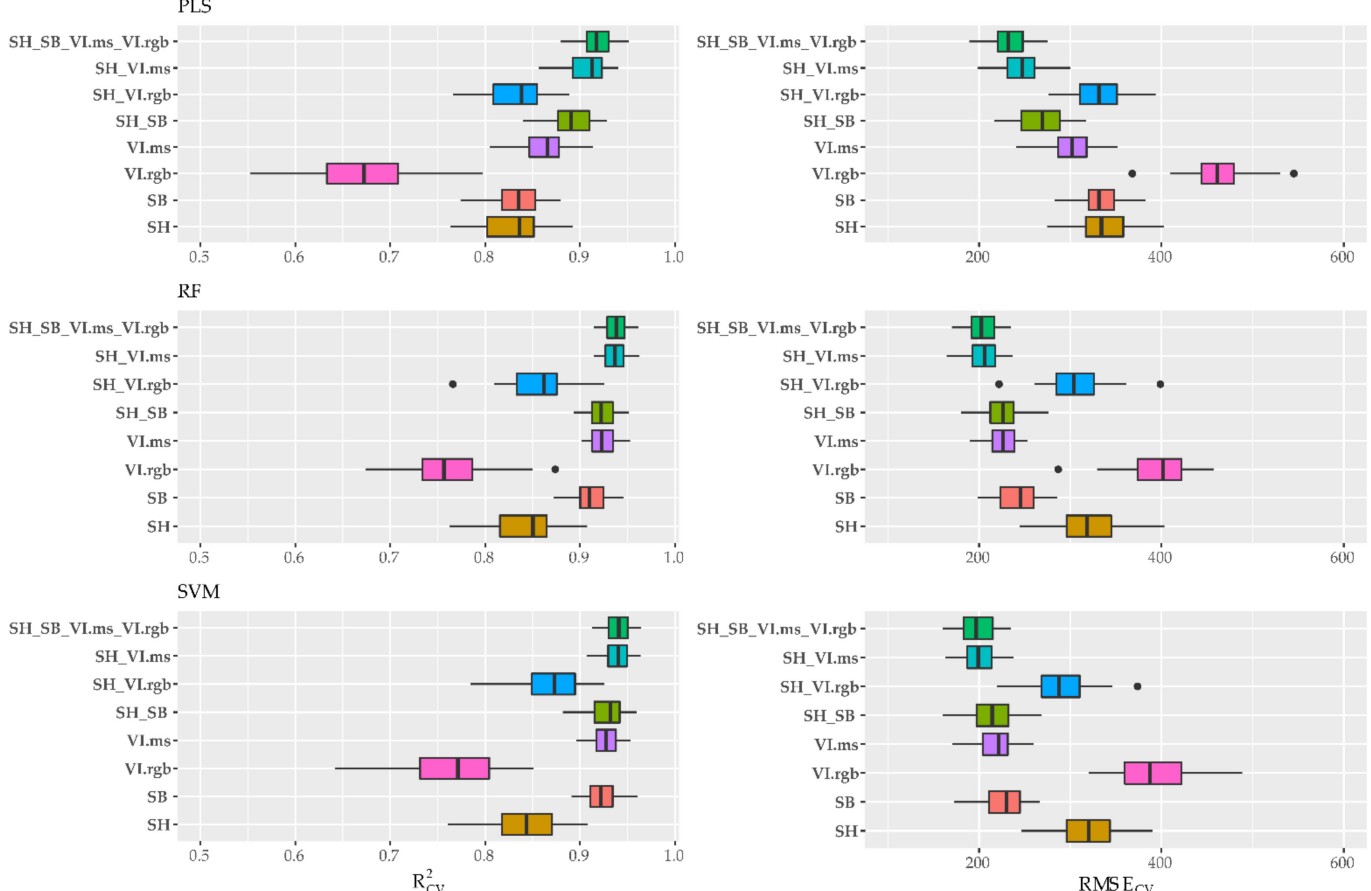

**Figure 12.** Box and whisker plot of $R^2_{CV}$ and RMSE$_{CV}$ (kg ha$^{-1}$) of all resamples of RF, SVM, and PLS models for DMY prediction (SH: sward height metrics. SB: single bands, VI.rgb: RGB indices, VI.ms: VNIR indices).

For the DMY estimation, the interquartile range (IQR) of the error distribution (RMSE$_{CV}$) was the lowest for the RF models. The best performing RF models were the feature combinations SH_VI.ms and SH_SB_VI.ms_VI.rgb (median RMSE$_{CV}$ 206 and 203 kg ha$^{-1}$, respectively, and IQR of 25 kg ha$^{-1}$ for both models). For the SVM models, the SH_VI.ms feature combination performed best (median RMSE$_{CV}$ 199 and IQR 27 kg ha$^{-1}$). The SH_SB models using SVM and RF performed well with a median RMSE$_{CV}$ of 214 and 226 kg ha$^{-1}$. Additionally, the VI.ms (RF and SVM) were among the best ten models ranked by RMSE. The highest error distribution was found using only VI.rgb with the SVM algorithm (IQR 62 kg ha$^{-1}$). Surprisingly, using only the single bands already lead to accurate results (median RMSE$_{CV}$ of 245 and 230 kg ha$^{-1}$ and IQR of 37 and 34 kg ha$^{-1}$ for RF and SVM, respectively). The best model by the PLS algorithm using all features combined ranked in between the previously mentioned SB models (median RMSE$_{CV}$ of 232 kg ha$^{-1}$ and IQR of 27 kg ha$^{-1}$). For a detailed listing of the performance metrics per model, the reader is referred to Table A2 in Appendix A.

Since it was assumed that the inclusion of the SH metrics significantly improved the prediction accuracy, the models with and without SH metrics were compared using the Wilcoxon signed-rank test (one-sided) based on the RMSE$_{CV}$ values. The assumption was

met in all instances. Additionally, the two best models per algorithm were compared (SH_SB_VI.ms_VI.rgb and SH_VI.ms). The inclusion of all variables improved the prediction accuracy for the PLS and RF models; however, for the SVM model, the prediction accuracy of the full model was not higher than for SH_VI.ms model (*p*-value: 0.157).

In Figure 13, the ten most important variables for the two best models per algorithm are presented (variable importance plots for all models, including all features for DMY estimation, are presented in Figures A1–A3 in Appendix A). The SH metrics were among the top variables for SVM and RF; whereas, for the PLS models, these features were less important. Except for the NGRDI in the full PLS model, no VI.rgb was among the top ten variables. In the RF models, MSAVI, GNDVI, NIR.RE, NDREI, and MSAVI were among the most important variables in both the full and SH_VI.ms models. In contrast, the ranking of the spectral variables in the SVM models was not as stable as in the RF models.

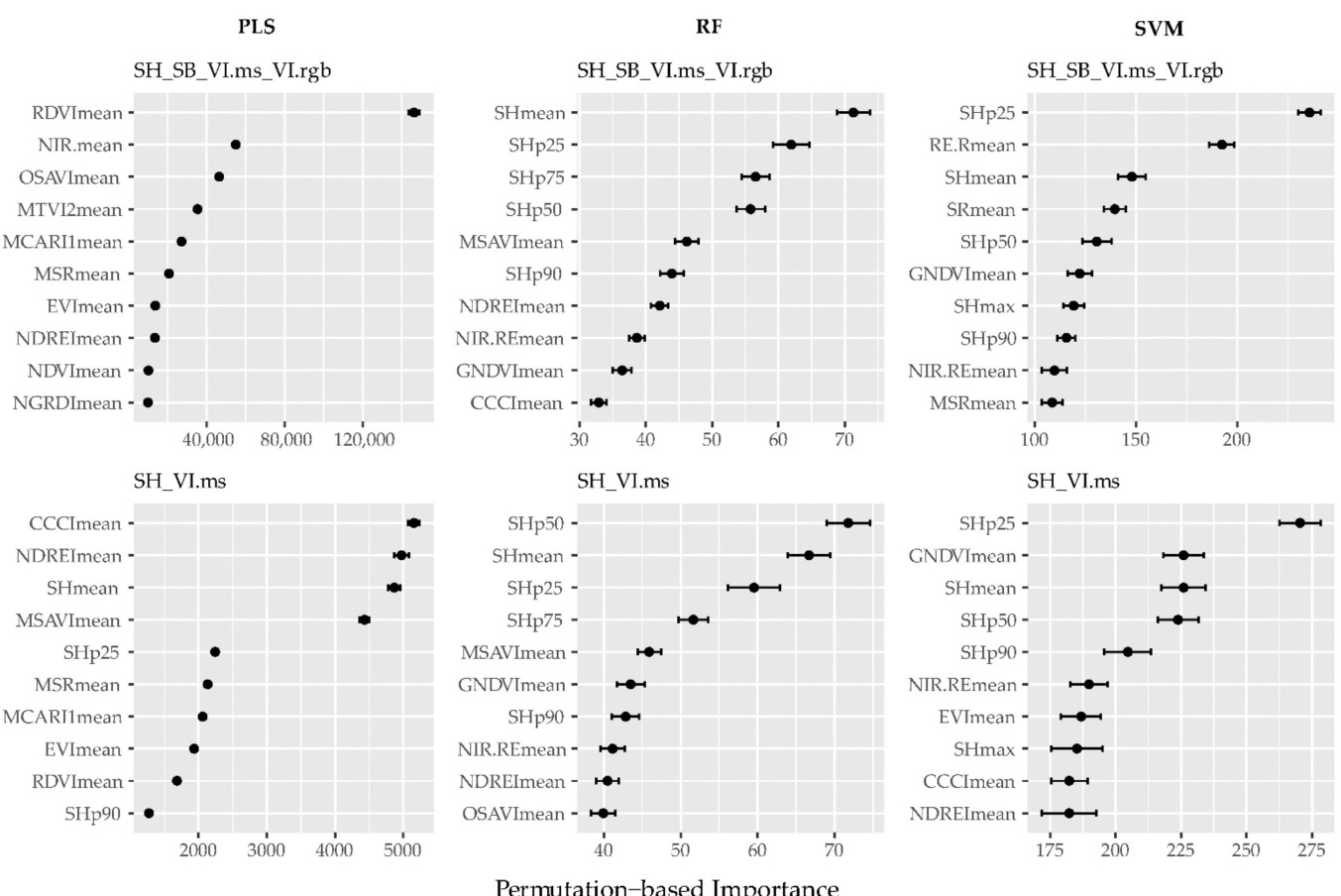

**Figure 13.** Variable importance plots of best models (10 most important variables shown) for DMY prediction. Variable importance is based on permutation.

### 3.5. N-Concentration Prediction

The results of all cross-validation model runs of the SLR for N% prediction are presented as box and whisker plots in Figure 14. No single predictor was able to adequately predict N%, and all models showed high variability in the cross-validation results. However, PPRI, a VI.rgb, showed the highest median $R^2_{CV}$ and lowest median $RMSE_{CV}$.

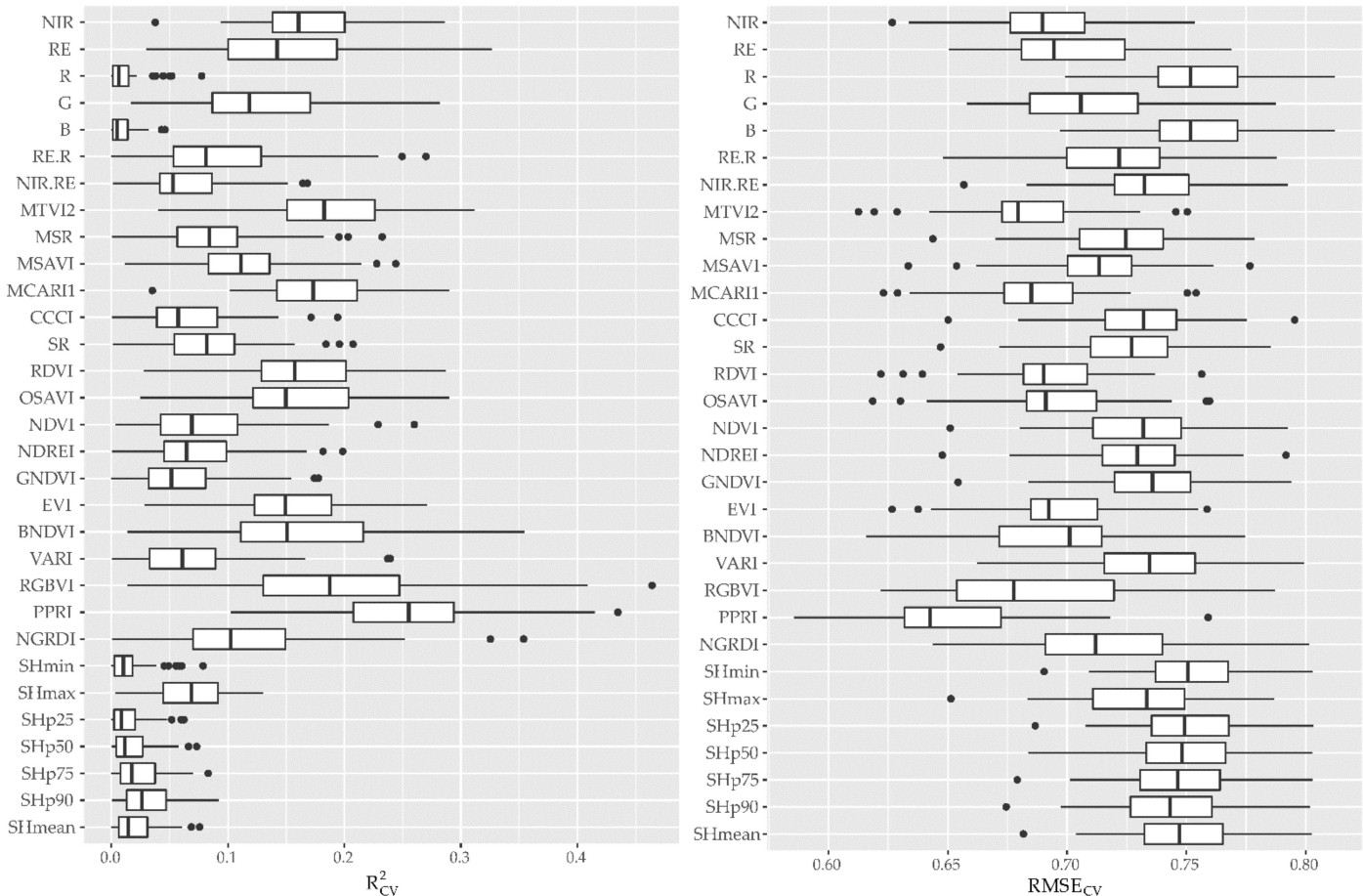

**Figure 14.** Box and whisker plot of $R^2_{CV}$ and $RMSE_{CV}$ (%) of all resamples of linear models (simple linear regression) for N-concentration prediction.

In Figure 15, the results of all cross-validation model runs of the three ML algorithms (PLS, SVM, RF) are shown based on different feature sets to predict N%. Irrespective of the algorithm, the SH and VI.rgb features and their combination provided the poorest results for N% estimation. Again, the PLS models generally exhibited poorer results than RF and SVM. The two best models were the combinations of all feature sets with a median $RMSE_{CV}$ of 0.31 and 0.32 N% for RF and SVM, respectively (both median $R^2_{CV}$ of 0.83). Surprisingly, the SH_SB combination yielded the third best result using the SVM algorithm with a median $R^2_{CV}$ of 0.81 and median $RMSE_{CV}$ of 0.32 N%. The SVM models using the VI.ms and SH_VI.ms feature sets showed similar performances to the SH_SB model. A detailed listing of the performance metrics can be found in Table A3 in Appendix A.

Similar to the DMY prediction models, it was tested whether the inclusion of SH metrics improved the prediction accuracy of N% using the Wilcoxon signed-rank test. In most cases, the assumption was met; only for the RF models using the single bands and the SVM models using the VI.ms was the difference not significant. The comparison between algorithms for the best models showed differences only significant to the PLS model (test results, see Table A5 in Appendix A).

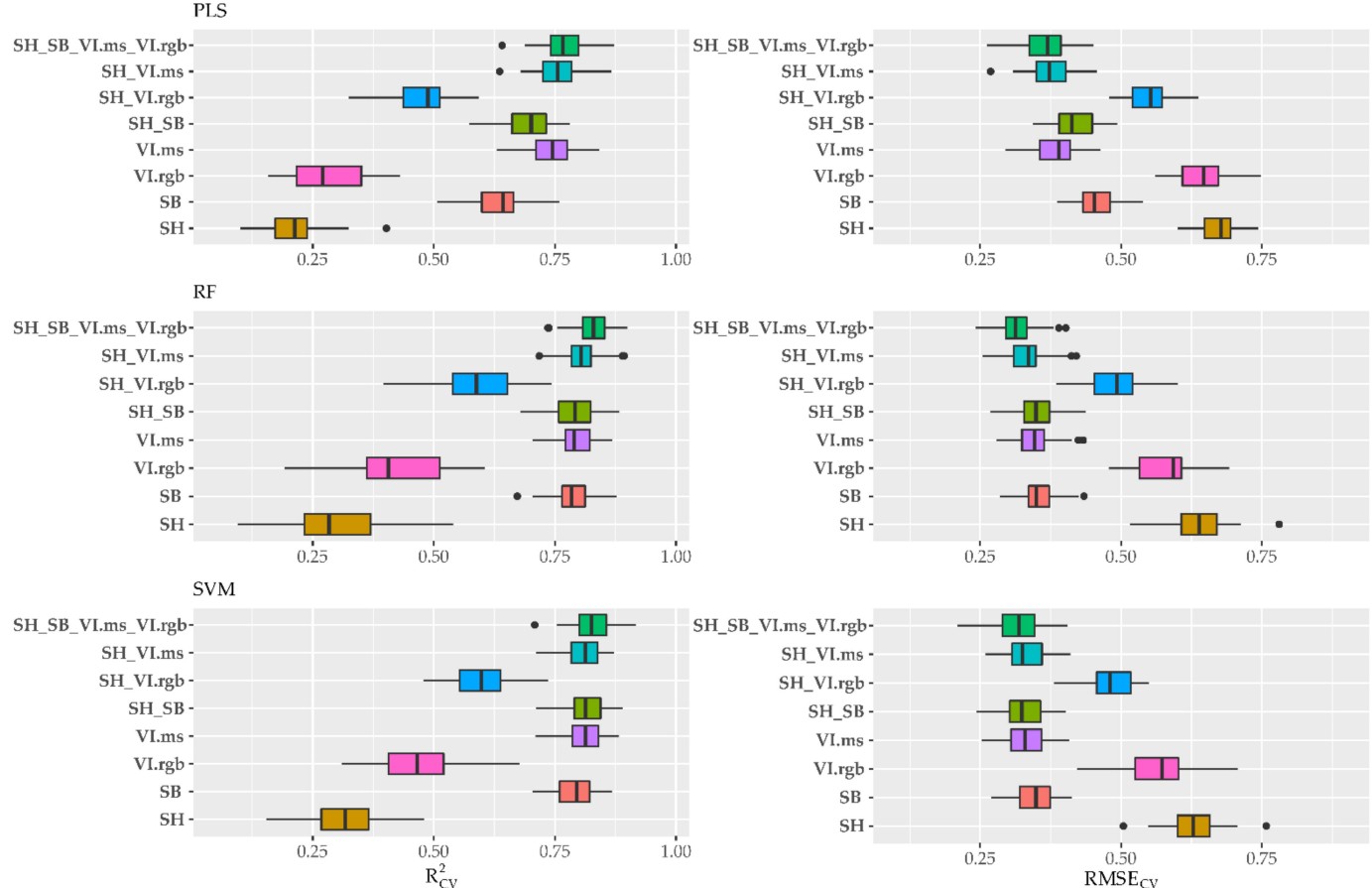

**Figure 15.** Box and whisker plot of $R^2_{CV}$ and $RMSE_{CV}$ (%) of all resamples of RF, SVM, and PLS models for N-concentration prediction (SH: sward height metrics. SB: single bands, VI.rgb: RGB-indices, VI.ms: VNIR indices).

In Figure 16, the variable importance plots of the two best models per algorithm are depicted. The SH features are only in the SVM models among the most important features. Common to most models are the important features, including the blue band and the NIR and red edge bands. The PPRI, an index from the visible region only, is the most important variable for the RF and SVM algorithms in the full model. Without VI.rgb included, the BNDVI is the paramount variable. Again, the most important variables in both best RF models are similar. Between the top variables of the SVM models, greater variation is present. Furthermore, the SVM model includes the RGBVI, similar to the PPRI, an index from the visible region, as the third best variable in the full model. For the PLS models mainly, the VI.ms are of higher importance than all SH, VI.rgb, or SB features. The VIPs for all models and all features can be found in Appendix A in Figures A4–A6 for completeness.

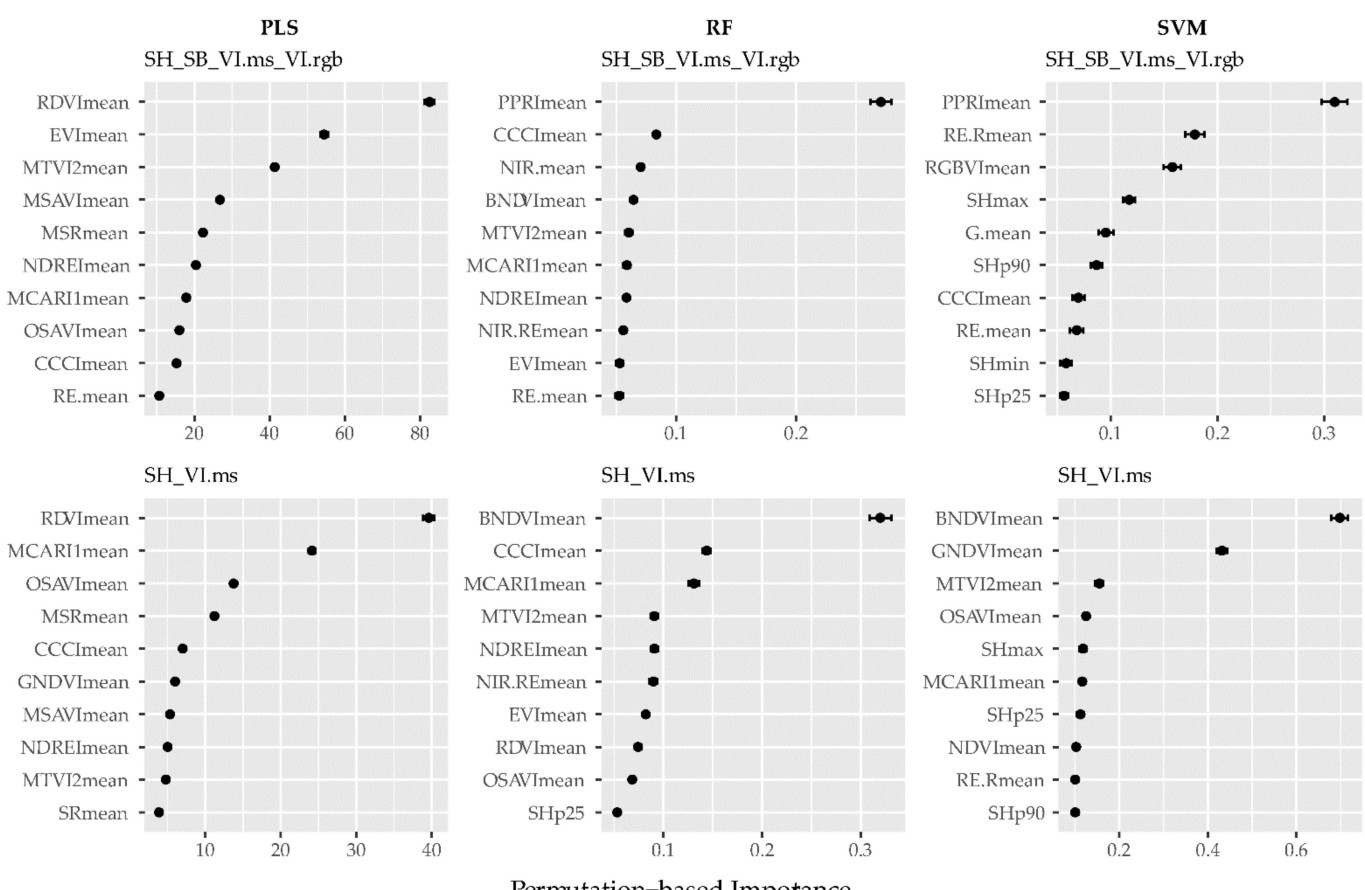

**Figure 16.** Variable importance plots of best models (10 most important variables shown) for N% prediction.

### 3.6. N Uptake Prediction

In Figure 17, the results of all cross-validation model runs of the SLR for N-uptake prediction are presented. The SH metrics mean, p90, p75, and p50 (median) show the best results, followed by GNDVI, NDREI, SR, CCCI, and the NIR/RE ratio with a median $R^2_{CV}$ of around 0.6. Except for the NIR band, VI.rgb and SB exhibit very low $R^2_{CV}$ (almost zero).

The results for the N uptake estimations of all cross-validation model runs of the three ML algorithms (PLS, SVM, RF) are shown based on different feature sets in Figure 18. In general, the PLS algorithm provides poorer results than the RF and SVM algorithms, and the VI.rgb feature set performs the least, irrespective of the modeling algorithm. Again, the best results could be achieved with the RF and SVM algorithms using all features or the SH_VI.ms combination (median $RMSE_{CV}$ of 7 kg N ha$^{-1}$ with IQR of 1 kg N ha$^{-1}$ for all four models). Furthermore, the SH_SB, and VI.ms models using RF or SVM performed similarly well. Compared to the linear models, the feature sets and combinations performed significantly better using the ML algorithms, except for the PLS-VI.rgb model. A detailed listing of the performance metrics for all N uptake estimation models can be found in Table A4 in Appendix A.

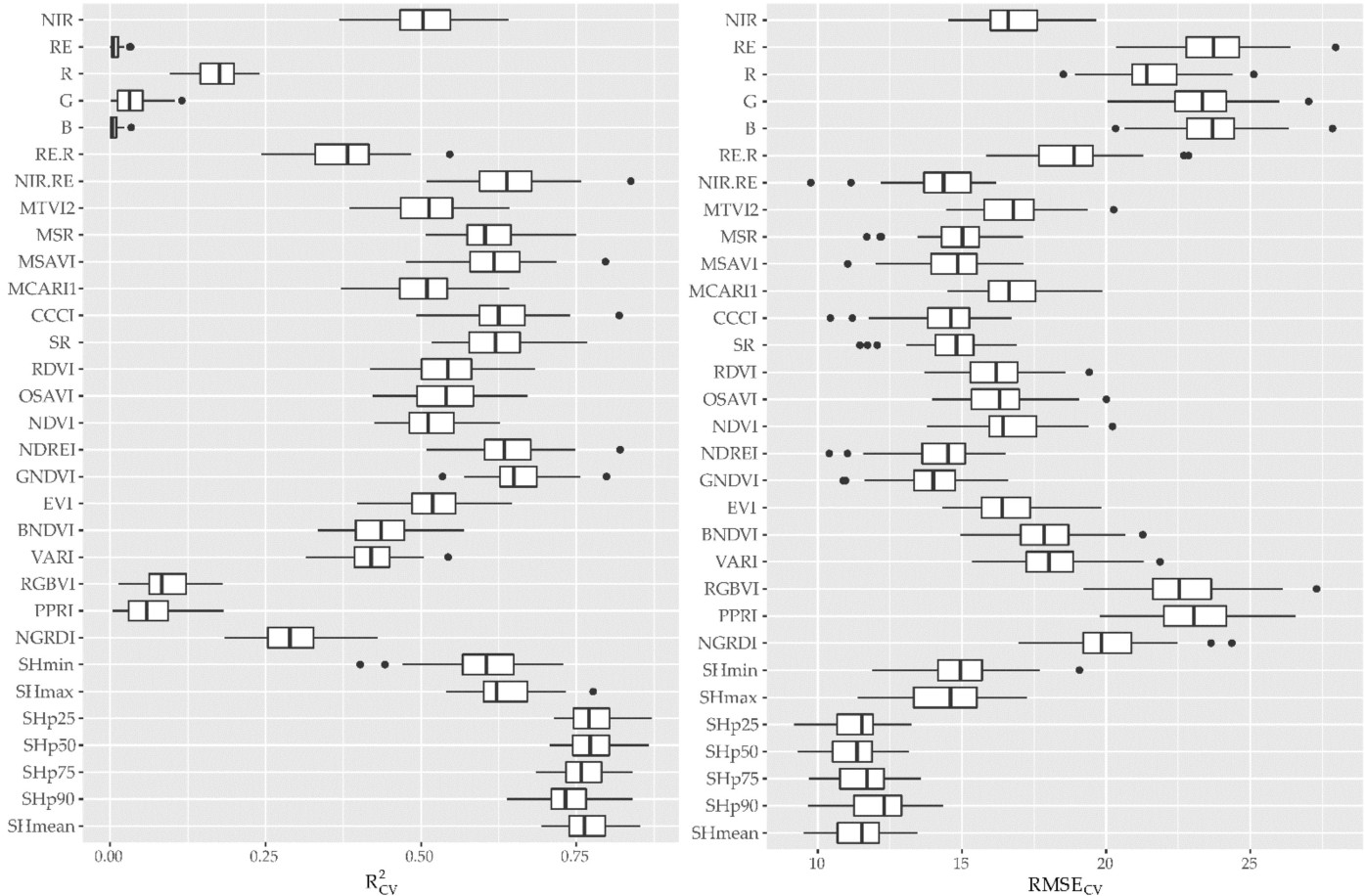

**Figure 17.** Box and whisker plot of $R^2_{CV}$ and $RMSE_{CV}$ (kg ha$^{-1}$) of all resamples of linear models (simple linear regression) for N-uptake prediction.

Again, the models were compared by the Wilcoxon signed-rank test to test the difference in prediction accuracy when SH metrics were included. The difference was significant for all combinations, except the RF SB model, so the inclusion of the SH metrics yielded no better result for N-uptake prediction with the single bands of the MS camera. The comparison between the two best models showed significant differences for the PLS and RF-based models, but was not significant between the RF and SVM models.

The variable importance plots (Figure 19) reveal that no clear pattern for Nup estimation can be distinguished between the best models of SVM and PLS. The SH metrics are of higher importance for the RF and SVM models than for the PLS models. In the RF models, the important features are comparable in their ranking, but no such stability can be achieved for the SVM models. Variable importance plots for all Nup estimation models, including all features, can be found in Figures A7–A9 in Appendix A for completeness.

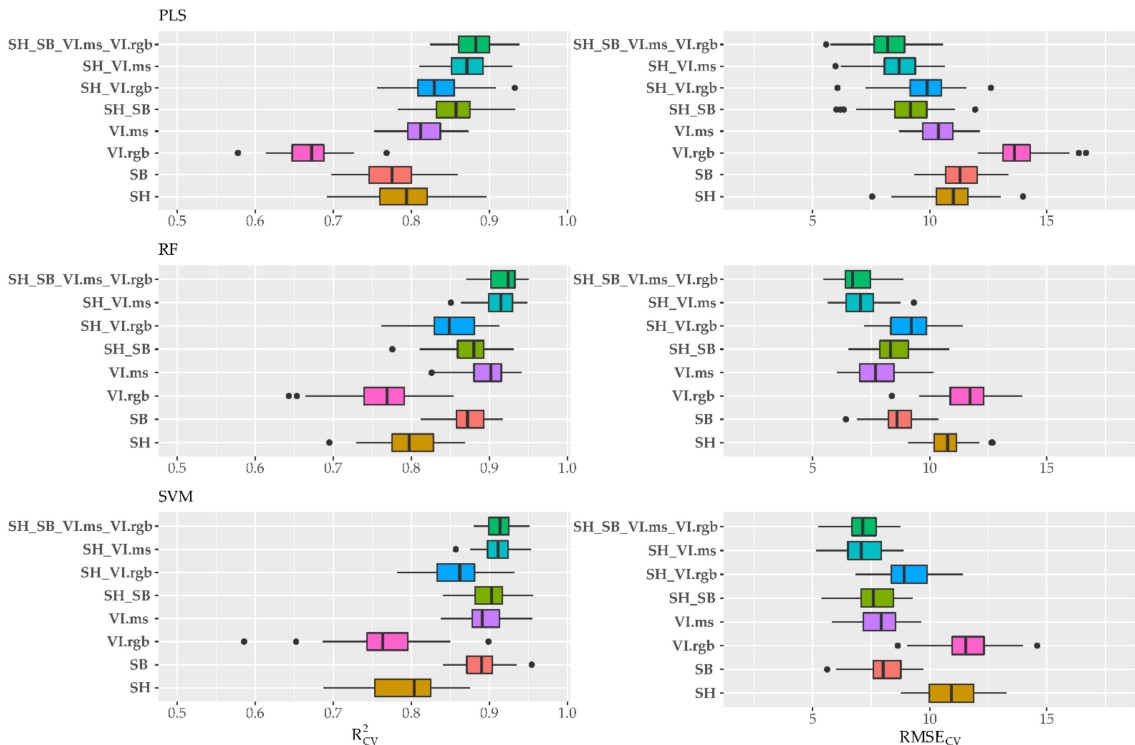

**Figure 18.** Box and whisker plot of $R^2_{CV}$ and $RMSE_{CV}$ (N uptake kg ha$^{-1}$) of all resamples of RF, SVM, and PLS models for N-uptake prediction (SH: sward height metrics. SB: single bands, VI.rgb: RGB indices, VI.ms: VNIR indices).

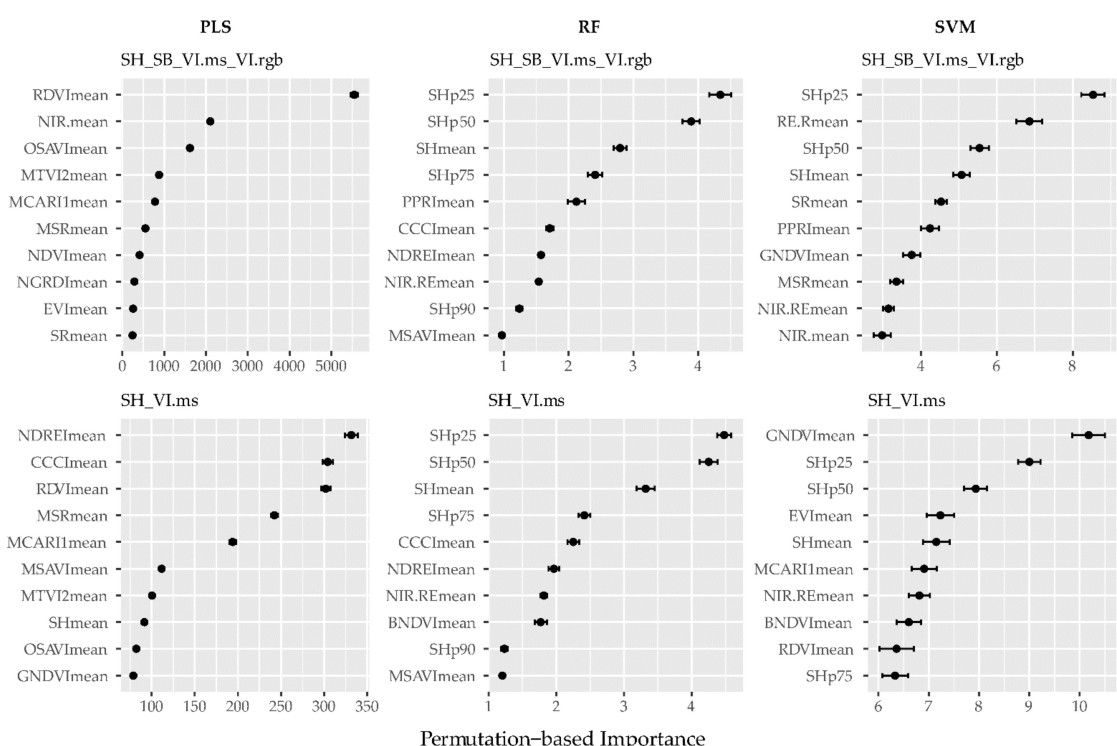

**Figure 19.** Variable importance plots of best models (10 most important variables shown) for N-uptake prediction.

### 3.7. Models with Reduced Features

Since it is desirable to implement models with as few features as possible, it was decided to select a set of important features based on the models developed in the previous sections. The performance of RF and SVM models utilizing all features was not significantly different; the RF algorithm was chosen to train new models, including only the ten most important variables based on the variable importance scores (see Figures 13, 16 and 19). The RMSE$_{CV}$ of the three models with reduced features for the estimation of DMY, N%, and Nup is shown in Figure 20. The respective median R$^2_{CV}$ values were 0.93, 0.80, and 0.91. The DMY estimation with reduced features resulted in a 5% loss of accuracy of median RMSE$_{CV}$ compared to the full model (203 kg ha$^{-1}$ compared to 213 kg ha$^{-1}$ DMY). For N% estimation, the loss of accuracy of median RMSE$_{CV}$ was higher (9%, 0.31 N% compared to 0.34 N%), but for Nup, the loss of accuracy was negligible (1.5%, 6.8 kg ha$^{-1}$ N compared to 7 kg ha$^{-1}$ N).

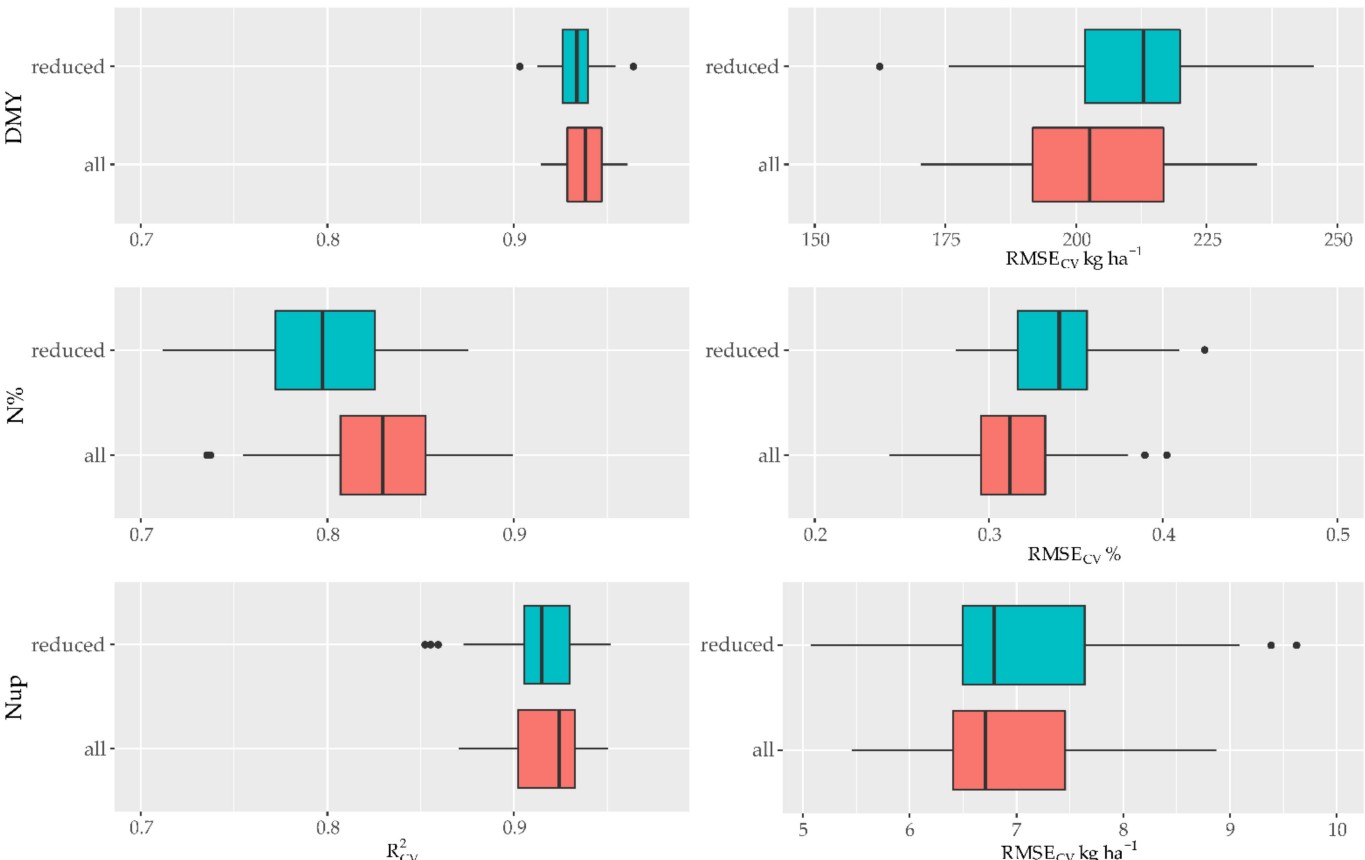

**Figure 20.** Box and whisker plot of RMSE$_{CV}$ of all resamples of the reduced feature sets and all features included (random forest) for DMY, N%, and Nup predictions. The ten best features based on variable importance of the full model are included in the reduced feature set.

### 4. Discussion

The primary aim of this study was to evaluate the suitability of combining UAV-based sward height metrics and vegetation indices to estimate DMY, N%, and Nup of mixed temperate grassland by comparing different feature combinations and machine learning models (PLS, RF, SVM). The models were evaluated by applying a 10-fold cross-validation procedure with five repeats to assess the variability in predictive accuracy. There are only a few studies investigating the combination of canopy height and spectral data to estimate DMY, N%, and Nup in grassland. However, the approach of combining structural and spectral data is not new at all. Earlier studies implemented, e.g., radar satellite data combined with hyperspectral satellite data for wheat monitoring [100], or combined LiDAR

and hyperspectral data for forest classification [101,102]. However, since the last decade, platforms, sensors, and algorithms are becoming more accessible to a broader scientific audience [43].

## 4.1. Data Accuracy

The conditions for data acquisition, such as sun angle, temperature, humidity, rain, wind speed, dust, and haze, are crucial for the accurate prediction of plant traits by UAV-based data. Furthermore, image quality can be influenced by flight speed, flying height, camera tilt, and exposure time. Ideally, these factors are stable through every flight mission and comparable for every mission. However, in practical farming and even under experimental conditions, these requirements are seldom met.

Although the MS sensor in this study was equipped with an irradiance sensor to correct for changes in incoming light, the parameters for multitemporal data acquisition should be either clear sky or fully overcast conditions to ensure consistent irradiance and no or only low wind speed to avoid moving canopies and ensure a stable flightpath. These instabilities are apparent in the radiometric assessment of the camera compared to the ASD readings using the additional gray panels for four sampling dates (Figure 10). The radiometric correction of the MS camera in the present study was based on the suggested settings of the official Agisoft Metashape workflow for MicaSense RedEdge cameras [103]. A custom calibration procedure, as suggested by [104], was therefore not applied. However, a thorough evaluation of the camera's spectral response with laboratory equipment, such as an integrating sphere or monochromator, preferably on a regular basis, is helpful in quantifying errors based on hardware degradation and standard radiometric correction workflow. This is especially important in light of real farming applications, where a shift in camera sensitivity could result in a severe loss of profit when a field is incorrectly characterized as unhealthy or healthy.

Regarding UAV-based structural features, in recent studies, it has been shown that UAV-based canopy height data can well replace RPM and other manual measurements and work well in swards above 30 cm height [44,45]. This was also one of the key findings of our study. However, a degree of uncertainty remains in the proposed method; in particular, the calculation of the base model generally seems to be influenced by the stubble height after the full harvest, as well as the remains of the harvest, and possible rodent activity. Furthermore, the calculation of a base model by interpolating bare soil points is not feasible for practical farming applications, since no paths between the plots were set up (in this case) and the sward was permanently managed for several years (even decades) without any plowing. Furthermore, calculating a base model by interpolating RTK-GPS points is rather laborious, especially when the site is not flat and shows small depressions and mounds. Therefore, our method was a compromise for practical farming applications, but further research should be directed to incorporate LiDAR data, especially for base model acquisition [105].

In addition, the timing of the sampling harvests was not ideal in every case. As mentioned in the Methods Section, lodging or heavily bent plots were excluded from the analysis, since they do not reflect the actual farming practice; swards are mown before reaching a certain maturity due to decreasing quality parameters, such as digestibility. Furthermore, the relationship between the studied parameters (DMY, N%, Nup) and UAV-based height or reflectance of the canopy is impaired when the plants are bent or even lodging [30,106]. Ultimately, a procedure, as proposed by Wilke et al. [107], could be integrated into DMY estimation models to correct for lodging severity.

The dry periods in the summer months were unusual compared to the long-term mean temperature and precipitation. Consequently, sward growth was negatively affected, and thus the third harvest in both years might not be representative of typical biomass values. This phenomenon might reflect the increasing prevalence of drought, warming, and climate variability [108]. Thus, the investigated years might therefore reflect changing yields in the future.

*4.2. Impact of Combining Structural and Spectral Data on Predictive Performance*

An important aspect of assessing biomass by means of spectral reflectance features is the well-known saturation effect of spectral reflectance with increasing biomass and leaf area index (LAI) [109]. Furthermore, the nutrient status of plants, especially N concentration, affects chlorophyll concentration in the leaves and therefore influences the reflectance characteristics [110–112]. These challenges can be addressed by integrating other biophysical plant traits into the modeling framework, such as plant or canopy height measured with various sensors. This has been observed for various crops [30,113] and grasslands [24,114].

Recently, comparable studies highlighted the potential of UAV-based data for trait estimation in grassland. In the study by Viljanen et al. [61], the best model utilized both feature types (derived from a UAV-mounted RGB and hyperspectral camera) using the RF algorithm, with an RMSE of 0.34 t ha$^{-1}$ for the DMY estimation for the primary growth. When structural and spectral features were combined, the CH features were of higher importance for the DMY prediction models than the spectral features. This was also observed in our study, except when using the PLS algorithm. Michez et al. [115] achieved an RMSE of 0.09 kg m$^2$ (900 kg ha$^{-1}$) for the DMY estimation by combining VIs and CH (adj. R$^2$ = 0.49), and also found that the CH had the highest relative importance in the multilinear regression model. Grüner et al. [55] investigated DMY and N fixation by comparing RF and PLS models of spectral features with and without texture features. They concluded that the additional texture features substantially improved the estimation models: the RMSE of the DMY estimation improved from 0.86 t ha$^{-1}$ to 0.52 t ha$^{-1}$ when texture features were included using RF. In the study of Karunaratne et al. [63], different feature combinations and flying heights were tested and the best models always considered both feature types (~400 kg ha$^{-1}$ RMSE). Similar to [63], a MicaSense RedEdge camera was employed in the study conducted by Pranga et al. [62]. Here, the combination of height and VIs yielded an RMSE of 382 kg ha$^{-1}$ DMY. Pranga et al. also found the CH features of highest importance when predicting DMY with a fused dataset (from the RGB camera as well as the MS camera); however, estimating the DMY with pure CH features yielded an rRMSE of 30–35%. In comparison, the rRMSE of the DMY estimation was 10% lower on average for the purely CH-based models in our study.

In a purely spectral approach, Togeiro de Alckmin et al. [116] achieved an RMSE of 397–464 kg ha$^{-1}$ depending on the modeling algorithm for the DMY estimation using a small MS camera (Parrot Sequoia) in a two-year study. Oliveira et al. [59] employed a similar sensor setup as [61] and reported an RMSE of 389 kg ha$^{-1}$ for the combination of CH fused with narrowband indices and single bands from an HS camera modeled by RF regression. In comparison, our study yielded significantly lower RMSE$_{CV}$ than all of the studies mentioned above. Depending on the algorithm, we achieved a median RMSE$_{CV}$ of around 200 kg ha$^{-1}$ DMY with a very narrow error range. However, the low RMSE might be affected by the range of the DMY samples, since both sampling years were affected by drought.

N concentration and uptake were also assessed by Oliveira et al. [59]. They found that the combination of structural (CH) and spectral (RGB and VNIR) features showed the best results (12.5% and 19% rRMSE, respectively, for the primary growth). This feature combination is comparable to our full model, although our study yielded an even lower median rRMSE of 10% for N% and 17% for Nup.

It needs to be noted here that N uptake is generally dependent on accurate measurements of both DMY and N%, since it is calculated as the product of both parameters. In the present experiment, the most important features for Nup were therefore the CH features and Vis, which also performed well in our N% estimation models. Predicting the N concentration by a single predictor is limited [109,117,118]. Therefore, it was not surprising that no single feature was able to adequately estimate N%. However, when using a set of features or a combination of structural and spectral features together with ML algorithms, the estimation accuracy rose significantly. The most important features for N%

estimation included the blue or red edge bands, like the PPRI, the BNDVI, the NDREI, the CCCI, and the NIR.RE ratio. This is in line with the literature, as these bands are sensitive to chlorophyll concentration in the leaves, which is closely, but not necessarily linear, linked to N [109,119–122].

A promising approach is integrating the SWIR spectral domain into biomass and N monitoring [34]. Such an application was successfully implemented by Jenal et al. [48,123] using selected spectral bands from a novel VNIR/SWIR imaging system. As our study showed, the information contained in the single MS camera bands fused with CH features yielded already accurate results. For DMY, N%, and Nup estimations, the performance of the SH_SB combination was in the same range as the computationally more expensive SH_VI.ms combination using RF or SVM. Thus, the selection of distinct wavelengths, which are more directly linked to, e.g., plant N, as in the SWIR domain, instead of using indirect links via chlorophyll absorption in the visible and red edge region, might reduce the need for VI calculation in an ML modeling framework. While hyperspectral sensors may outperform MS sensors, the latter are easier to operate and investment costs are substantially lower.

*4.3. Transferability and Generality of Models*

The practical benefit of prediction models based on spatially explicit data lies in the possibility to derive maps of management zones depicting the desired feature, such as N uptake or DMY. These maps can aid in decision making for site-specific treatments (e.g., fertilizer, pest control), optimal time of harvest for silage and hay production, or grazing management, e.g., in combination with virtual fencing technology [124]. However, the empirically derived models need to generalize well and have to be transferable to different sites and years.

We hypothesized that combining UAV-based structural and spectral data with ML modeling could build adequate models to estimate important grassland management parameters, such as DMY, N%, and Nup. By incorporating different years with various weather and climate conditions, applying different N treatments and having a significant slope in our field experiment, we accounted for the substantial variations in our dataset. Thus, the data are able to cover a considerable amount of naturally occurring differences in the biomass and N status of grasslands in temperate regions with similar management practices. As discussed by Geipel et al. [58], the transferability and generalizability of their models increased when calibration was performed using the pooled dataset. This was also observed in our models. Models built with a dataset based on multiple years, and preferably on multiple sites, are able to generalize better due to the higher variation of the dataset, and thus may reflect conditions on other sites and during different years better. However, ML algorithms are known for their excellent performance of in-sample predictions, but poorer performance on unseen data [125,126]. Thus, our hypothesis has yet to be proven by transferring the derived models to unseen data, e.g., new sites.

## 5. Conclusions and Outlook

Monitoring biomass and N-related traits is crucial for the sustainable and profitable management decisions in grasslands. This study investigated the potential of UAV-based structural and spectral features and their combination to predict DMY, N%, and Nup. Models built with structural features (SH metrics) from a consumer-grade RGB camera and spectral features from a small MS camera using linear regression and three ML algorithms (PLS, RF, SVM) were compared. Overall, models with a combination of both structural and spectral features improved the prediction of DMY, N%, and Nup, regardless of the ML algorithm. However, the PLS models were outperformed by their RF and SVM counterparts. The best models for DMY estimation were the full models using all features (SH_SB_VI.ms_VI.rgb) and the combination SH_VI.ms using the RF or SVM algorithms, with a median RMSE of 197–206 kg ha$^{-1}$. Similarly, Nup was best estimated with the aforementioned feature combinations by RF and SVM (median RMSE 7 kg ha$^{-1}$). N% was best

estimated using the full model or the combinations of SH_SB and SH_VI.ms (median RMSE of 0.31–0.33%). Surprisingly, the combination of single bands of the MS camera and SH features yielded already accurate results for all three traits. This observation implies that the accurate modeling of DMY, N%, and Nup could be achieved with less computational effort, since the VI calculation could be omitted. Furthermore, the performance of the models with a reduced feature set based on feature selection (best ten features using RF) was also only marginally less than the best models. Thus, the estimations of the three important grassland parameters, DMY, N%, and Nup, are easily achievable with UAV-based data from relatively inexpensive, small imaging sensors and ML models. The presented approaches may offer alternatives to traditional destructive measurements of important plant traits and improve estimation accuracy. These spatially explicit predictions can aid in a range of agricultural applications for site-specific management, such as yield maps. Although the transferability of our models needs to be validated with independent datasets (e.g., different sites and years), this study demonstrated the applicability of UAV-based structural and spectral features for accurate DMY, N%, and Nup estimations. Future research should be directed towards integrating LiDAR data, especially for the acquisition of an accurate base model for CH calculation. Additionally, the integration of texture features seems to be a promising approach, especially for biomass estimation. Furthermore, the short-wave infrared region would be a beneficial addition to the models, due to the direct link to plant N.

This study expanded on the current research of the combination of structural and spectral features from UAV-based imaging sensors to estimate the important traits in temperate grasslands using machine learning algorithms. All in all, our study demonstrated the potential of UAV-based data for agricultural applications, and highlighted areas for future research using remote sensing for grassland management.

**Author Contributions:** Conceptualization, U.L., M.L.G., J.S. and G.B.; data curation, U.L., A.B. and M.L.G.; formal analysis, U.L., I.K. and M.L.G.; funding acquisition, J.J., J.S. and G.B.; investigation, U.L., A.B. and M.L.G.; methodology, U.L., A.B., M.L.G. and G.B.; supervision, J.J., J.S. and G.B.; writing—original draft, U.L. and G.B.; writing—review and editing, U.L., A.B., I.K., J.J., M.L.G., J.S. and G.B. All authors have read and agreed to the published version of the manuscript.

**Funding:** This research was partly funded by the German Federal Ministry of Education and Research (BMBF) (Grant number: 031B0734F), as part of the consortium research project "Green-Grass". We acknowledge support for the Article Processing Charge from the DFG (German Research Foundation, 491454339).

**Data Availability Statement:** The data presented in this study are available on request from the corresponding author.

**Acknowledgments:** We would like to thank the local farmer Reinhard Mosler and our student staff Jannis Menne, Joscha Grasshoff, Lilian Bromen, Marina Herbrecht, Christoph Müller, Nikolai Kirch, Mirijam Zickel, and Mika Thuning. Furthermore, we would like to thank Michael Schmidt from Arbeitskreis Landwirtschaft, Wasser und Boden (ALWB) in Siegburg. Last, but not least, we would like to thank Hubert Hüging from Bonn University.

**Conflicts of Interest:** The authors declare no conflict of interest.

## Appendix A

**Table A1.** Cross-validation Regression results simple linear regression for all features.

| Feature | DMY kg ha$^{-1}$ | | | | N % Biomass | | | | N up kg ha$^{-1}$ | | | |
|---|---|---|---|---|---|---|---|---|---|---|---|---|
| | $R^2$ | sd | RMSE | sd | $R^2$ | sd | RMSE | sd | $R^2$ | sd | RMSE | sd |
| SHmean | 0.82 ± 0.04 | | 342 ± 31 | | 0.02 ± 0.02 | | 0.75 ± 0.02 | | 0.77 ± 0.04 | | 11.5 ± 1.0 | |
| SHp90 | 0.82 ± 0.04 | | 341 ± 32 | | 0.03 ± 0.02 | | 0.74 ± 0.03 | | 0.74 ± 0.04 | | 12.2 ± 1.1 | |
| SHp75 | 0.83 ± 0.04 | | 337 ± 30 | | 0.02 ± 0.02 | | 0.75 ± 0.02 | | 0.76 ± 0.04 | | 11.6 ± 1.0 | |
| SHp50 | 0.82 ± 0.04 | | 342 ± 31 | | 0.02 ± 0.02 | | 0.75 ± 0.02 | | 0.78 ± 0.04 | | 11.3 ± 0.9 | |
| SHp25 | 0.81 ± 0.04 | | 357 ± 32 | | 0.02 ± 0.02 | | 0.75 ± 0.02 | | 0.77 ± 0.04 | | 11.3 ± 0.9 | |
| SHmax | 0.77 ± 0.04 | | 391 ± 33 | | 0.07 ± 0.03 | | 0.73 ± 0.03 | | 0.63 ± 0.05 | | 14.4 ± 1.3 | |
| SHmin | 0.62 ± 0.06 | | 497 ± 36 | | 0.02 ± 0.02 | | 0.75 ± 0.02 | | 0.61 ± 0.07 | | 14.9 ± 1.5 | |
| NGRDI | 0.44 ± 0.05 | | 606 ± 40 | | 0.11 ± 0.07 | | 0.72 ± 0.03 | | 0.29 ± 0.05 | | 20.1 ± 1.5 | |
| PPRI | 0.01 ± 0.01 | | 807 ± 28 | | 0.26 ± 0.07 | | 0.65 ± 0.03 | | 0.07 ± 0.05 | | 23.1 ± 1.6 | |
| RGBVI | 0.22 ± 0.05 | | 717 ± 40 | | 0.20 ± 0.10 | | 0.69 ± 0.04 | | 0.09 ± 0.04 | | 22.7 ± 1.6 | |
| VARI | 0.55 ± 0.05 | | 546 ± 36 | | 0.07 ± 0.05 | | 0.73 ± 0.03 | | 0.42 ± 0.05 | | 18.2 ± 1.4 | |
| BNDVI | 0.64 ± 0.03 | | 485 ± 25 | | 0.16 ± 0.08 | | 0.70 ± 0.03 | | 0.44 ± 0.06 | | 17.9 ± 1.2 | |
| EVI | 0.70 ± 0.03 | | 445 ± 21 | | 0.16 ± 0.05 | | 0.70 ± 0.03 | | 0.52 ± 0.06 | | 16.5 ± 1.3 | |
| GNDVI | 0.79 ± 0.02 | | 370 ± 23 | | 0.06 ± 0.04 | | 0.73 ± 0.03 | | 0.65 ± 0.05 | | 14.1 ± 1.3 | |
| NDREI | 0.79 ± 0.03 | | 374 ± 24 | | 0.07 ± 0.04 | | 0.73 ± 0.03 | | 0.64 ± 0.06 | | 14.3 ± 1.3 | |
| NDVI | 0.67 ± 0.04 | | 470 ± 31 | | 0.08 ± 0.06 | | 0.73 ± 0.03 | | 0.52 ± 0.05 | | 16.7 ± 1.3 | |
| OSAVI | 0.74 ± 0.02 | | 415 ± 20 | | 0.16 ± 0.06 | | 0.69 ± 0.03 | | 0.54 ± 0.06 | | 16.2 ± 1.3 | |
| RDVI | 0.74 ± 0.03 | | 416 ± 20 | | 0.16 ± 0.05 | | 0.69 ± 0.03 | | 0.54 ± 0.06 | | 16.1 ± 1.3 | |
| SR | 0.78 ± 0.03 | | 384 ± 28 | | 0.08 ± 0.05 | | 0.73 ± 0.03 | | 0.62 ± 0.06 | | 14.7 ± 1.2 | |
| CCCI | 0.77 ± 0.03 | | 389 ± 24 | | 0.07 ± 0.04 | | 0.73 ± 0.03 | | 0.63 ± 0.06 | | 14.5 ± 1.3 | |
| MCARI1 | 0.70 ± 0.03 | | 447 ± 22 | | 0.18 ± 0.05 | | 0.69 ± 0.03 | | 0.51 ± 0.06 | | 16.7 ± 1.3 | |
| MSAVI | 0.79 ± 0.03 | | 369 ± 23 | | 0.11 ± 0.05 | | 0.71 ± 0.03 | | 0.62 ± 0.06 | | 14.7 ± 1.3 | |
| MSR | 0.78 ± 0.03 | | 383 ± 27 | | 0.09 ± 0.05 | | 0.72 ± 0.03 | | 0.61 ± 0.05 | | 14.9 ± 1.2 | |
| MTVI2 | 0.72 ± 0.03 | | 431 ± 20 | | 0.19 ± 0.05 | | 0.68 ± 0.03 | | 0.51 ± 0.06 | | 16.6 ± 1.3 | |
| NIR.RE | 0.76 ± 0.03 | | 397 ± 26 | | 0.06 ± 0.04 | | 0.73 ± 0.03 | | 0.64 ± 0.06 | | 14.3 ± 1.3 | |
| RE.R | 0.53 ± 0.07 | | 554 ± 47 | | 0.10 ± 0.06 | | 0.72 ± 0.03 | | 0.38 ± 0.06 | | 18.8 ± 1.5 | |
| B | 0.01 ± 0.02 | | 805 ± 31 | | 0.01 ± 0.01 | | 0.75 ± 0.02 | | 0.01 ± 0.01 | | 23.7 ± 1.6 | |
| G | 0.02 ± 0.02 | | 804 ± 30 | | 0.13 ± 0.06 | | 0.71 ± 0.03 | | 0.04 ± 0.03 | | 23.4 ± 1.6 | |
| R | 0.20 ± 0.05 | | 726 ± 39 | | 0.01 ± 0.02 | | 0.75 ± 0.02 | | 0.18 ± 0.03 | | 21.6 ± 1.4 | |
| RE | 0.02 ± 0.02 | | 804 ± 29 | | 0.15 ± 0.06 | | 0.70 ± 0.03 | | 0.01 ± 0.01 | | 23.7 ± 1.6 | |
| NIR | 0.69 ± 0.03 | | 455 ± 24 | | 0.17 ± 0.05 | | 0.69 ± 0.03 | | 0.51 ± 0.06 | | 16.7 ± 1.2 | |

**Table A2.** Cross-validation results for DMY estimation using PLS, RF and SVM regression (nRMSE normalized via mean observed value). Bold highlights indicate two best models per algorithm. iqr: interquartile range, sd: standard deviation.

| | Modelname | Hyperparameters | R² Median | | iqr | Mean | | sd | RMSE kg ha⁻¹ Median | | iqr | Mean | | sd | nRMSE % Median | | iqr | Mean | | sd |
|---|---|---|---|---|---|---|---|---|---|---|---|---|---|---|---|---|---|---|---|---|
| **PLS** | SH | ncomp = 3 | 0.84 | ± | 0.05 | 0.83 | ± | 0.04 | 334 | ± | 41 | 336 | ± | 31 | 25.1 | ± | 3.1 | 25.2 | ± | 2.3 |
| | SB | ncomp = 4 | 0.84 | ± | 0.04 | 0.83 | ± | 0.02 | 331 | ± | 28 | 331 | ± | 23 | 24.9 | ± | 2.1 | 24.9 | ± | 1.7 |
| | VI.rgb | ncomp = 3 | 0.67 | ± | 0.08 | 0.67 | ± | 0.05 | 461 | ± | 35 | 465 | ± | 35 | 34.6 | ± | 2.6 | 34.9 | ± | 2.6 |
| | VI.ms | ncomp = 14 | 0.87 | ± | 0.03 | 0.86 | ± | 0.02 | 302 | ± | 31 | 302 | ± | 24 | 22.7 | ± | 2.3 | 22.6 | ± | 1.8 |
| | SH_SB | ncomp = 11 | 0.89 | ± | 0.03 | 0.89 | ± | 0.02 | 269 | ± | 43 | 269 | ± | 25 | 20.2 | ± | 3.2 | 20.2 | ± | 1.9 |
| | SH_VI.rgb | ncomp = 5 | 0.84 | ± | 0.05 | 0.83 | ± | 0.03 | 331 | ± | 41 | 333 | ± | 30 | 24.9 | ± | 3.1 | 25.0 | ± | 2.2 |
| | **SH_VI.ms** | **ncomp = 18** | **0.91** | ± | **0.03** | **0.91** | ± | **0.02** | **247** | ± | **30** | **247** | ± | **24** | **18.6** | ± | **2.2** | **18.6** | ± | **1.8** |
| | **SH_SB_VI.ms_VI.rgb** | **ncomp = 29** | **0.92** | ± | **0.02** | **0.92** | ± | **0.02** | **232** | ± | **27** | **232** | ± | **21** | **17.4** | ± | **2.0** | **17.5** | ± | **1.6** |
| **RF** | SH | mtry = 2 | 0.85 | ± | 0.05 | 0.84 | ± | 0.03 | 319 | ± | 49 | 320 | ± | 32 | 23.9 | ± | 3.7 | 24.0 | ± | 2.4 |
| | SB | mtry = 4 | 0.91 | ± | 0.02 | 0.91 | ± | 0.02 | 245 | ± | 37 | 242 | ± | 22 | 18.4 | ± | 2.8 | 18.2 | ± | 1.7 |
| | VI.rgb | mtry = 2 | 0.76 | ± | 0.05 | 0.76 | ± | 0.04 | 402 | ± | 48 | 395 | ± | 37 | 30.2 | ± | 3.6 | 29.7 | ± | 2.8 |
| | VI.ms | mtry = 15 | 0.92 | ± | 0.02 | 0.92 | ± | 0.01 | 226 | ± | 24 | 225 | ± | 18 | 17.0 | ± | 1.8 | 16.9 | ± | 1.3 |
| | SH_SB | mtry = 12 | 0.92 | ± | 0.02 | 0.92 | ± | 0.01 | 226 | ± | 26 | 227 | ± | 21 | 17.0 | ± | 1.9 | 17.1 | ± | 1.5 |
| | SH_VI.rgb | mtry = 3 | 0.86 | ± | 0.04 | 0.86 | ± | 0.03 | 304 | ± | 41 | 306 | ± | 32 | 22.8 | ± | 3.1 | 23.0 | ± | 2.4 |
| | **SH_VI.ms** | **mtry = 8** | **0.94** | ± | **0.02** | **0.94** | ± | **0.01** | **206** | ± | **25** | **205** | ± | **17** | **15.5** | ± | **1.9** | **15.4** | ± | **1.3** |
| | **SH_SB_VI.ms_VI.rgb** | **mtry = 12** | **0.94** | ± | **0.02** | **0.94** | ± | **0.01** | **203** | ± | **25** | **203** | ± | **17** | **15.2** | ± | **1.9** | **15.2** | ± | **1.3** |
| **SVM** | SH | C = 0.5, sigma = 1.62 | 0.84 | ± | 0.05 | 0.84 | ± | 0.03 | 320 | ± | 48 | 320 | ± | 32 | 24.1 | ± | 3.6 | 24.0 | ± | 2.4 |
| | SB | C = 4, sigma = 0.62 | 0.92 | ± | 0.02 | 0.92 | ± | 0.02 | 230 | ± | 34 | 226 | ± | 22 | 17.3 | ± | 2.6 | 17.0 | ± | 1.6 |
| | VI.rgb | C = 4, sigma = 0.95 | 0.77 | ± | 0.07 | 0.77 | ± | 0.05 | 388 | ± | 62 | 393 | ± | 42 | 29.1 | ± | 4.7 | 29.5 | ± | 3.2 |
| | VI.ms | C = 4, sigma = 0.24 | 0.93 | ± | 0.02 | 0.93 | ± | 0.02 | 221 | ± | 27 | 219 | ± | 21 | 16.6 | ± | 2.0 | 16.4 | ± | 1.6 |
| | SH_SB | C = 4, sigma = 0.24 | 0.93 | ± | 0.03 | 0.93 | ± | 0.02 | 214 | ± | 35 | 215 | ± | 24 | 16.1 | ± | 2.6 | 16.2 | ± | 1.8 |
| | SH_VI.rgb | C = 4, sigma = 0.27 | 0.87 | ± | 0.05 | 0.87 | ± | 0.03 | 288 | ± | 42 | 290 | ± | 33 | 21.6 | ± | 3.1 | 21.8 | ± | 2.5 |
| | **SH_VI.ms** | **C = 2, sigma = 0.16** | **0.94** | ± | **0.02** | **0.94** | ± | **0.01** | **199** | ± | **27** | **200** | ± | **19** | **14.9** | ± | **2.0** | **15.0** | ± | **1.4** |
| | **SH_SB_VI.ms_VI.rgb** | **C = 4, sigma = 0.07** | **0.94** | ± | **0.02** | **0.94** | ± | **0.01** | **197** | ± | **32** | **198** | ± | **20** | **14.8** | ± | **2.4** | **14.9** | ± | **1.5** |

**Table A3.** Cross-validation results for N% estimation using PLS, RF and SVM regression (nRMSE normalized via mean observed value). Bold highlights indicate two best models per algorithm. iqr: interquartile range, sd: standard deviation.

| | Modelname | Hyperparameters | R² Median | iqr | Mean | sd | RMSE N % Median | iqr | Mean | sd | nRMSE % Median | iqr | Mean | sd |
|---|---|---|---|---|---|---|---|---|---|---|---|---|---|---|
| **PLS** | SH | ncomp = 3 | 0.21 ± | 0.07 | 0.21 ± | 0.06 | 0.68 ± | 0.05 | 0.67 ± | 0.03 | 22.6 ± | 1.5 | 22.4 ± | 1.1 |
| | SB | ncomp = 4 | 0.64 ± | 0.07 | 0.63 ± | 0.05 | 0.45 ± | 0.05 | 0.46 ± | 0.04 | 15.1 ± | 1.6 | 15.3 ± | 1.2 |
| | VI.rgb | ncomp = 3 | 0.27 ± | 0.13 | 0.28 ± | 0.08 | 0.65 ± | 0.06 | 0.64 ± | 0.04 | 21.5 ± | 2.1 | 21.4 ± | 1.5 |
| | VI.ms | ncomp = 14 | 0.75 ± | 0.06 | 0.74 ± | 0.05 | 0.39 ± | 0.05 | 0.39 ± | 0.04 | 13.0 ± | 1.8 | 12.9 ± | 1.3 |
| | SH_SB | ncomp = 10 | 0.70 ± | 0.07 | 0.70 ± | 0.05 | 0.41 ± | 0.06 | 0.42 ± | 0.04 | 13.7 ± | 2.0 | 13.9 ± | 1.2 |
| | SH_VI.rgb | ncomp = 10 | 0.49 ± | 0.08 | 0.47 ± | 0.06 | 0.55 ± | 0.05 | 0.55 ± | 0.04 | 18.4 ± | 1.8 | 18.3 ± | 1.3 |
| | **SH_VI.ms** | **ncomp = 21** | **0.76** ± | **0.06** | **0.75** ± | **0.05** | **0.37** ± | **0.05** | **0.38** ± | **0.04** | **12.4** ± | **1.7** | **12.5** ± | **1.3** |
| | **SH_SB_VI.ms_VI.rgb** | **ncomp = 30** | **0.77** ± | **0.06** | **0.76** ± | **0.05** | **0.37** ± | **0.06** | **0.37** ± | **0.04** | **12.3** ± | **1.9** | **12.3** ± | **1.3** |
| **RF** | SH | mtry = 7 | 0.28 ± | 0.14 | 0.30 ± | 0.09 | 0.64 ± | 0.06 | 0.63 ± | 0.05 | 21.3 ± | 2.1 | 21.2 ± | 1.7 |
| | SB | mtry = 5 | 0.78 ± | 0.05 | 0.78 ± | 0.04 | 0.35 ± | 0.04 | 0.35 ± | 0.03 | 11.7 ± | 1.2 | 11.8 ± | 1.0 |
| | VI.rgb | mtry = 2 | 0.41 ± | 0.15 | 0.43 ± | 0.10 | 0.59 ± | 0.07 | 0.58 ± | 0.05 | 19.8 ± | 2.5 | 19.2 ± | 1.8 |
| | VI.ms | mtry = 15 | 0.79 ± | 0.05 | 0.79 ± | 0.04 | 0.35 ± | 0.04 | 0.35 ± | 0.03 | 11.6 ± | 1.3 | 11.6 ± | 1.1 |
| | SH_SB | mtry = 12 | 0.79 ± | 0.07 | 0.79 ± | 0.05 | 0.35 ± | 0.05 | 0.35 ± | 0.03 | 11.6 ± | 1.5 | 11.7 ± | 1.1 |
| | SH_VI.rgb | mtry = 11 | 0.59 ± | 0.11 | 0.59 ± | 0.07 | 0.49 ± | 0.07 | 0.49 ± | 0.04 | 16.4 ± | 2.2 | 16.3 ± | 1.5 |
| | **SH_VI.ms** | **mtry = 22** | **0.80** ± | **0.04** | **0.81** ± | **0.04** | **0.34** ± | **0.04** | **0.34** ± | **0.04** | **11.2** ± | **1.3** | **11.2** ± | **1.2** |
| | **SH_SB_VI.ms_VI.rgb** | **mtry = 30** | **0.83** ± | **0.05** | **0.83** ± | **0.04** | **0.31** ± | **0.04** | **0.32** ± | **0.03** | **10.4** ± | **1.2** | **10.5** ± | **1.1** |
| **SVM** | SH | C = 0.5, sigma = 1.80 | 0.32 ± | 0.10 | 0.32 ± | 0.08 | 0.63 ± | 0.06 | 0.63 ± | 0.05 | 20.9 ± | 1.9 | 21.0 ± | 1.6 |
| | SB | C = 4, sigma = 0.51 | 0.79 ± | 0.06 | 0.79 ± | 0.04 | 0.35 ± | 0.05 | 0.35 ± | 0.04 | 11.6 ± | 1.8 | 11.5 ± | 1.2 |
| | VI.rgb | C = 4, sigma = 0.89 | 0.47 ± | 0.11 | 0.47 ± | 0.09 | 0.57 ± | 0.08 | 0.57 ± | 0.06 | 19.1 ± | 2.6 | 18.9 ± | 2.1 |
| | VI.ms | C = 4, sigma = 0.27 | 0.81 ± | 0.05 | 0.81 ± | 0.04 | 0.33 ± | 0.05 | 0.33 ± | 0.04 | 11.0 ± | 1.8 | 11.0 ± | 1.3 |
| | **SH_SB** | C = 4, sigma = 0.19 | **0.81** ± | **0.05** | **0.81** ± | **0.04** | **0.32** ± | **0.05** | **0.33** ± | **0.04** | **10.8** ± | **1.8** | **10.9** ± | **1.2** |
| | SH_VI.rgb | C = 4, sigma = 0.31 | 0.60 ± | 0.08 | 0.60 ± | 0.06 | 0.48 ± | 0.06 | 0.48 ± | 0.04 | 16.0 ± | 2.0 | 16.0 ± | 1.5 |
| | **SH_VI.ms** | **C = 4, sigma = 0.17** | **0.81** ± | **0.05** | **0.81** ± | **0.04** | **0.32** ± | **0.05** | **0.33** ± | **0.04** | **10.8** ± | **1.8** | **11.0** ± | **1.2** |
| | **SH_SB_VI.ms_VI.rgb** | **C = 4, sigma = 0.07** | **0.83** ± | **0.06** | **0.83** ± | **0.04** | **0.32** ± | **0.06** | **0.32** ± | **0.04** | **10.6** ± | **1.9** | **10.6** ± | **1.3** |

**Table A4.** Cross-validation results for Nup estimation using PLS, RF and SVM regression (nRMSE normalized via mean observed value). Bold highlights indicate two best models per algorithm. iqr: interquartile range, sd: standard deviation.

| | Modelname | Hyperparameters | $R^2$ Median | iqr | Mean | sd | RMSE Median | iqr | Mean | sd | nRMSE Median | iqr | Mean | sd |
|---|---|---|---|---|---|---|---|---|---|---|---|---|---|---|
| **PLS** | SH | ncomp = 3 | 0.79 | ± 0.06 | 0.79 | ± 0.05 | 11.0 | ± 1.3 | 10.9 | ± 1.2 | 28.6 | ± 3.5 | 28.3 | ± 3.1 |
| | SB | ncomp = 4 | 0.78 | ± 0.05 | 0.77 | ± 0.04 | 11.3 | ± 1.3 | 11.4 | ± 0.9 | 29.4 | ± 3.5 | 29.7 | ± 2.5 |
| | VI.rgb | ncomp = 3 | 0.67 | ± 0.04 | 0.67 | ± 0.04 | 13.6 | ± 1.2 | 13.8 | ± 1.1 | 35.5 | ± 3.0 | 36.0 | ± 2.8 |
| | VI.ms | ncomp = 14 | 0.81 | ± 0.04 | 0.81 | ± 0.03 | 10.4 | ± 1.3 | 10.4 | ± 0.9 | 27.0 | ± 3.4 | 27.0 | ± 2.4 |
| | SH_SB | ncomp = 9 | 0.86 | ± 0.04 | 0.86 | ± 0.04 | 9.2 | ± 1.4 | 9.1 | ± 1.2 | 23.9 | ± 3.6 | 23.6 | ± 3.2 |
| | SH_VI.rgb | ncomp = 7 | 0.83 | ± 0.05 | 0.83 | ± 0.04 | 9.9 | ± 1.3 | 9.8 | ± 1.2 | 25.8 | ± 3.4 | 25.5 | ± 3.2 |
| | **SH_VI.ms** | **ncomp = 20** | **0.87** | ± **0.04** | **0.87** | ± **0.03** | **8.7** | ± **1.3** | **8.6** | ± **1.1** | **22.6** | ± **3.4** | **22.4** | ± **2.8** |
| | **SH_SB_VI.ms_VI.rgb** | **ncomp = 29** | **0.88** | ± **0.04** | **0.88** | ± **0.03** | **8.2** | ± **1.3** | **8.2** | ± **1.1** | **21.4** | ± **3.4** | **21.3** | ± **2.8** |
| **RF** | SH | mtry = 7 | 0.80 | ± 0.05 | 0.80 | ± 0.04 | 10.8 | ± 0.9 | 10.7 | ± 0.8 | 28.0 | ± 2.5 | 27.9 | ± 2.1 |
| | SB | mtry = 5 | 0.87 | ± 0.03 | 0.87 | ± 0.03 | 8.6 | ± 1.0 | 8.7 | ± 0.9 | 22.4 | ± 2.5 | 22.5 | ± 2.4 |
| | VI.rgb | mtry = 2 | 0.77 | ± 0.05 | 0.76 | ± 0.05 | 11.7 | ± 1.4 | 11.6 | ± 1.2 | 30.5 | ± 3.7 | 30.2 | ± 3.0 |
| | VI.ms | mtry = 15 | 0.90 | ± 0.03 | 0.89 | ± 0.03 | 7.7 | ± 1.5 | 7.8 | ± 1.0 | 20.0 | ± 3.8 | 20.3 | ± 2.6 |
| | SH_SB | mtry = 12 | 0.88 | ± 0.03 | 0.88 | ± 0.03 | 8.3 | ± 1.2 | 8.4 | ± 0.9 | 21.7 | ± 3.1 | 22.0 | ± 2.3 |
| | SH_VI.rgb | mtry = 10 | 0.85 | ± 0.05 | 0.85 | ± 0.03 | 9.2 | ± 1.5 | 9.2 | ± 0.9 | 24.0 | ± 4.0 | 24.0 | ± 2.5 |
| | **SH_VI.ms** | **mtry = 21** | **0.91** | ± **0.03** | **0.91** | ± **0.02** | **7.0** | ± **1.1** | **7.1** | ± **0.9** | **18.3** | ± **3.0** | **18.5** | ± **2.3** |
| | **SH_SB_VI.ms_VI.rgb** | **mtry = 27** | **0.92** | ± **0.03** | **0.92** | ± **0.02** | **6.7** | ± **1.1** | **6.9** | ± **0.8** | **17.5** | ± **2.7** | **17.9** | ± **2.1** |
| **SVM** | SH | C = 1, sigma = 1.62 | 0.80 | ± 0.07 | 0.79 | ± 0.05 | 10.9 | ± 1.9 | 10.9 | ± 1.2 | 28.4 | ± 4.9 | 28.4 | ± 3.0 |
| | SB | C = 4, sigma = 0.62 | 0.89 | ± 0.03 | 0.89 | ± 0.02 | 8.0 | ± 1.2 | 8.1 | ± 0.9 | 20.8 | ± 3.1 | 21.0 | ± 2.4 |
| | VI.rgb | C = 4, sigma = 0.95 | 0.76 | ± 0.05 | 0.77 | ± 0.05 | 11.5 | ± 1.4 | 11.6 | ± 1.2 | 30.0 | ± 3.6 | 30.1 | ± 3.1 |
| | VI.ms | C = 4, sigma = 0.24 | 0.89 | ± 0.03 | 0.89 | ± 0.02 | 7.9 | ± 1.4 | 7.9 | ± 0.9 | 20.7 | ± 3.6 | 20.6 | ± 2.4 |
| | SH_SB | C = 4, sigma = 0.24 | 0.90 | ± 0.04 | 0.90 | ± 0.03 | 7.6 | ± 1.4 | 7.6 | ± 1.0 | 19.8 | ± 3.6 | 19.9 | ± 2.5 |
| | SH_VI.rgb | C = 2, sigma = 0.27 | 0.86 | ± 0.05 | 0.86 | ± 0.04 | 8.9 | ± 1.5 | 9.0 | ± 1.1 | 23.2 | ± 3.9 | 23.5 | ± 2.9 |
| | **SH_VI.ms** | **C = 4, sigma = 0.16** | **0.91** | ± **0.03** | **0.91** | ± **0.02** | **7.1** | ± **1.4** | **7.1** | ± **0.9** | **18.4** | ± **3.7** | **18.6** | ± **2.4** |
| | **SH_SB_VI.ms_VI.rgb** | **C = 4, sigma = 0.07** | **0.91** | ± **0.03** | **0.91** | ± **0.02** | **7.1** | ± **1.0** | **7.1** | ± **0.8** | **18.6** | ± **2.7** | **18.5** | ± **2.2** |

**Table A5.** Results Wilcoxon signed-rank test.

| | Wilcoxon Signed Rank Test | | Significance Level adj. *p*-Value | | |
|---|---|---|---|---|---|
| | **Model 1** | **Model 2** | **DMY** | **N%** | **Nup** |
| PLS | VI.ms | SH_VI.ms | **** | **** | **** |
| | SB | SH_SB | **** | **** | **** |
| | VI.rgb | SH_VI.rgb | **** | **** | **** |
| | SH_VI.ms | SH_SB_VI.ms_VI.rgb | **** | **** | **** |
| RF | VI.ms | SH_VI.ms | **** | *** | **** |
| | SB | SH_SB | **** | ns | ns |
| | VI.rgb | SH_VI.rgb | **** | **** | **** |
| | SH_VI.ms | SH_SB_VI.ms_VI.rgb | ** | **** | **** |
| SVM | VI.ms | SH_VI.ms | **** | ns | **** |
| | SB | SH_SB | *** | **** | ** |
| | VI.rgb | SH_VI.rgb | **** | **** | **** |
| | SH_VI.ms | SH_SB_VI.ms_VI.rgb | ns | **** | ns |

Significance levels of adjusted *p*-values: **** $\leq$ 0.0001, *** $\leq$ 0.001, ** $\leq$ 0.01, ns > 0.05.

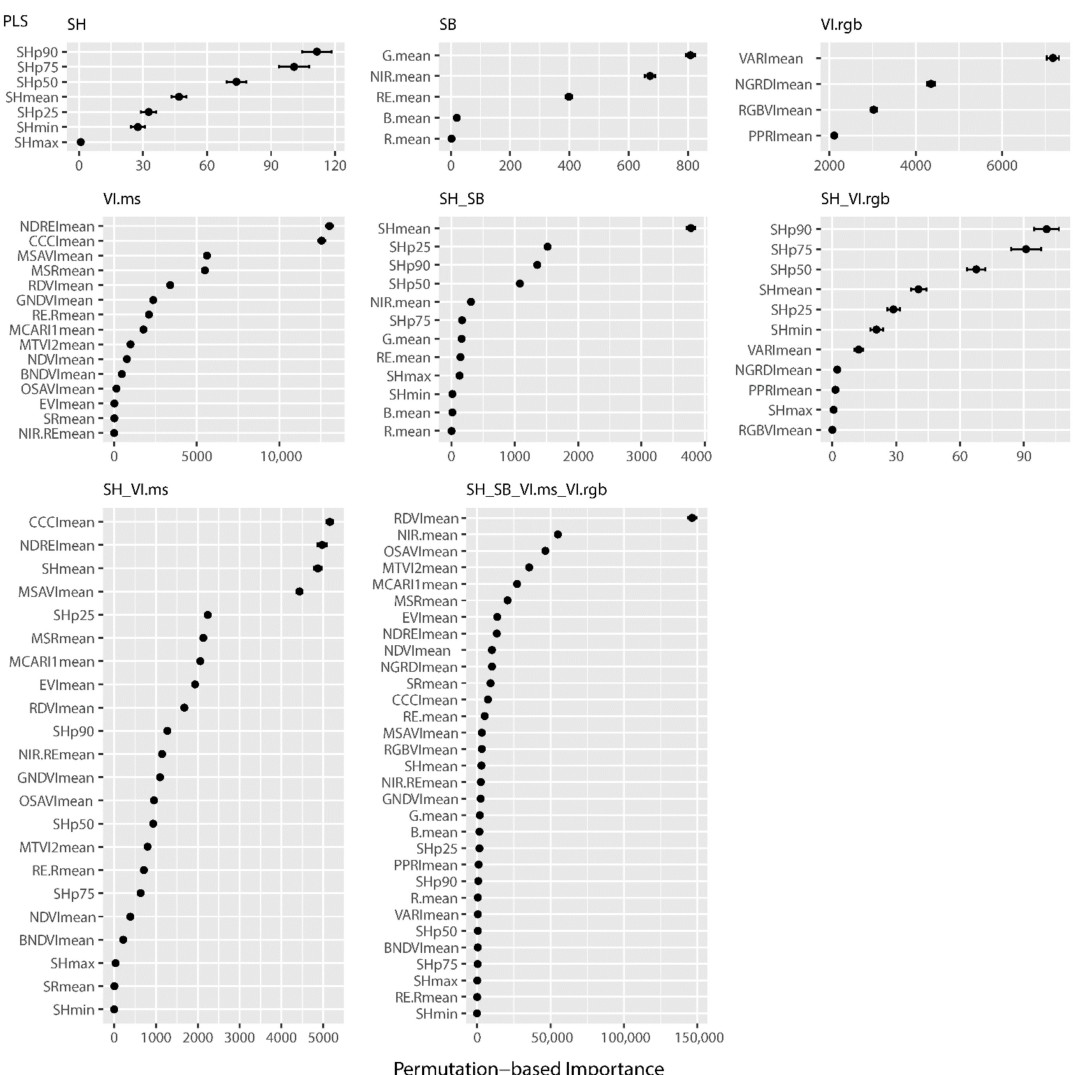

**Figure A1.** VIP of DMY estimation with PLS.

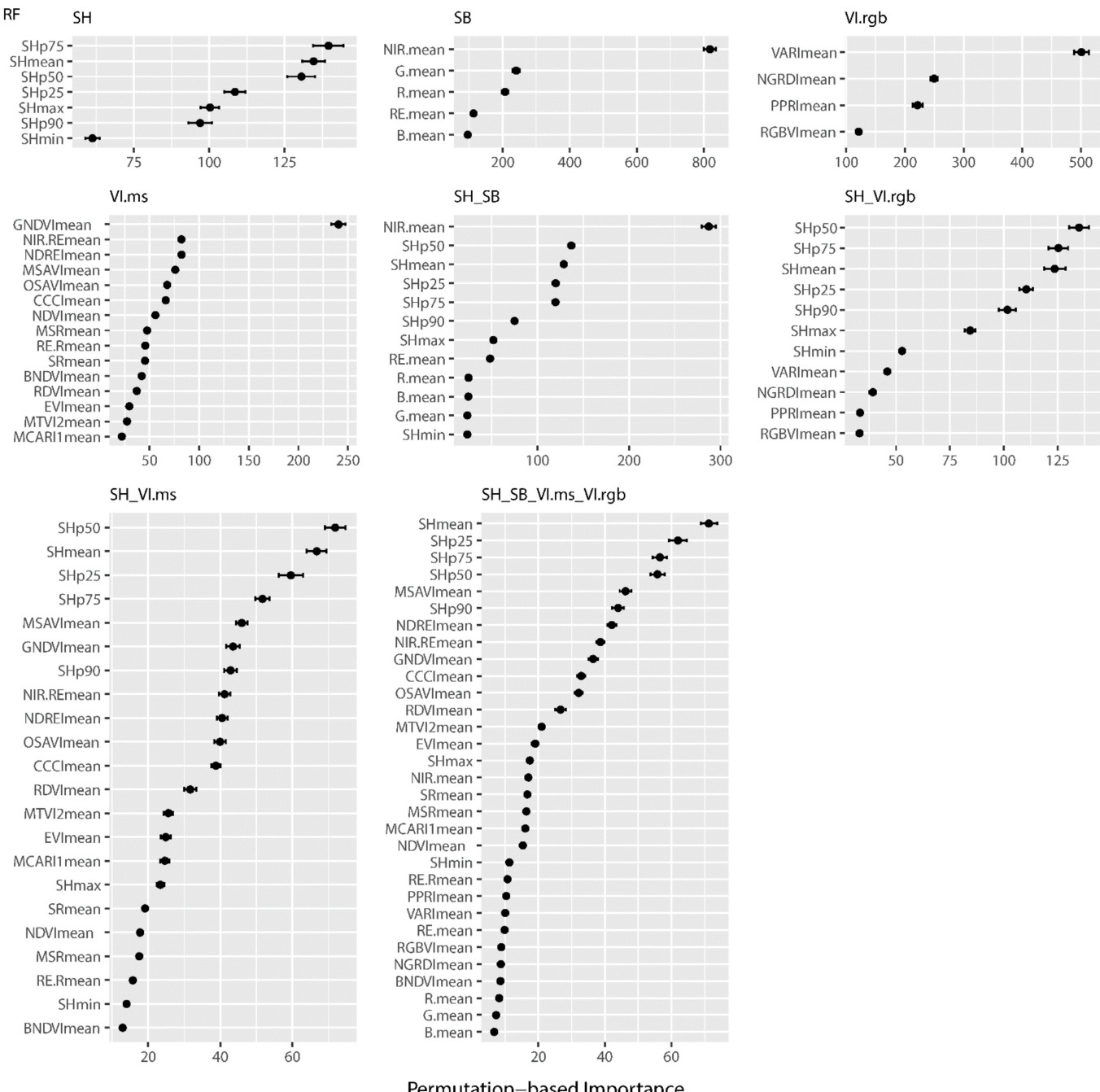

**Figure A2.** VIP of DMY estimation with RF.

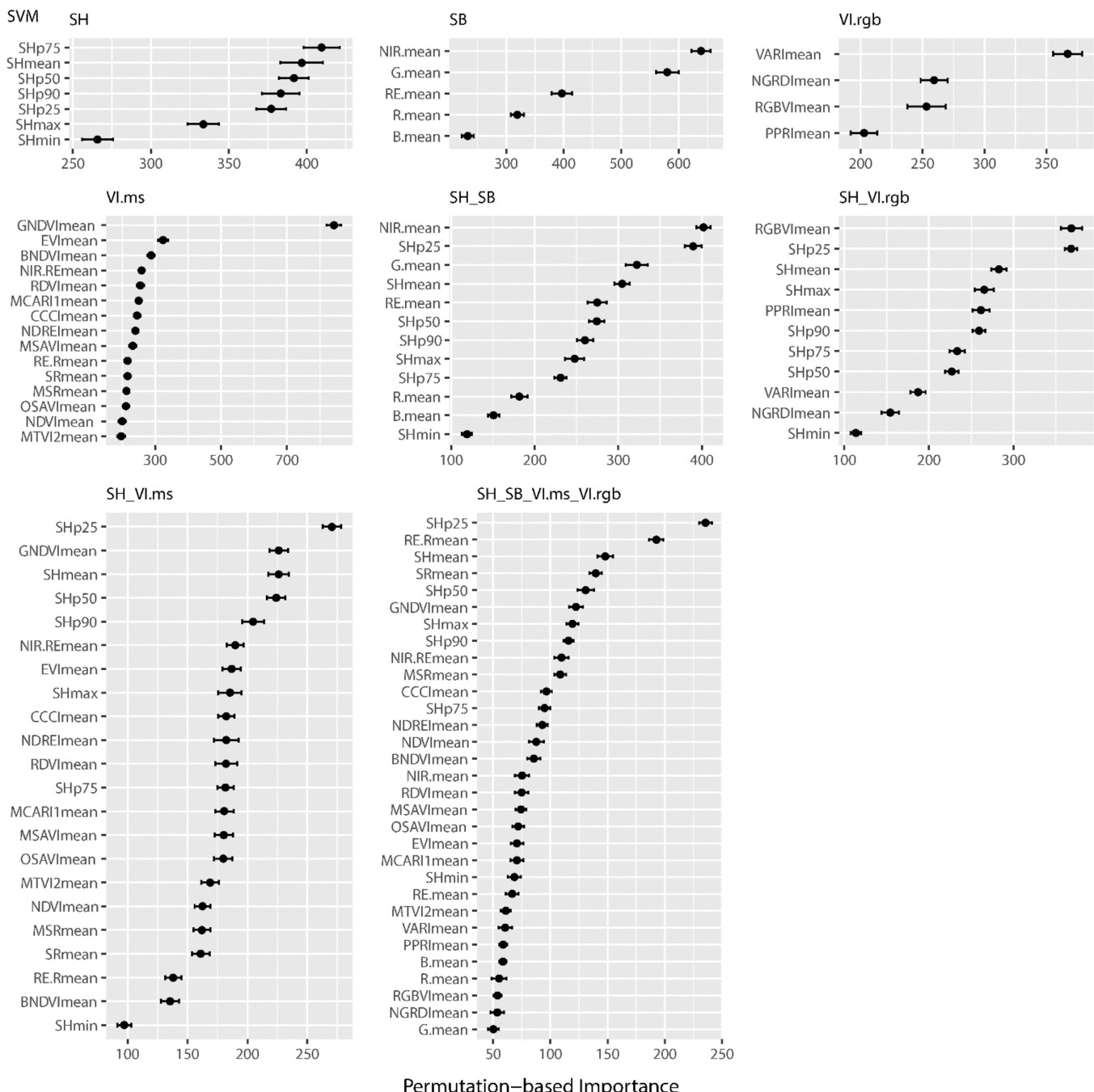

**Figure A3.** VIP of DMY estimation with SVM.

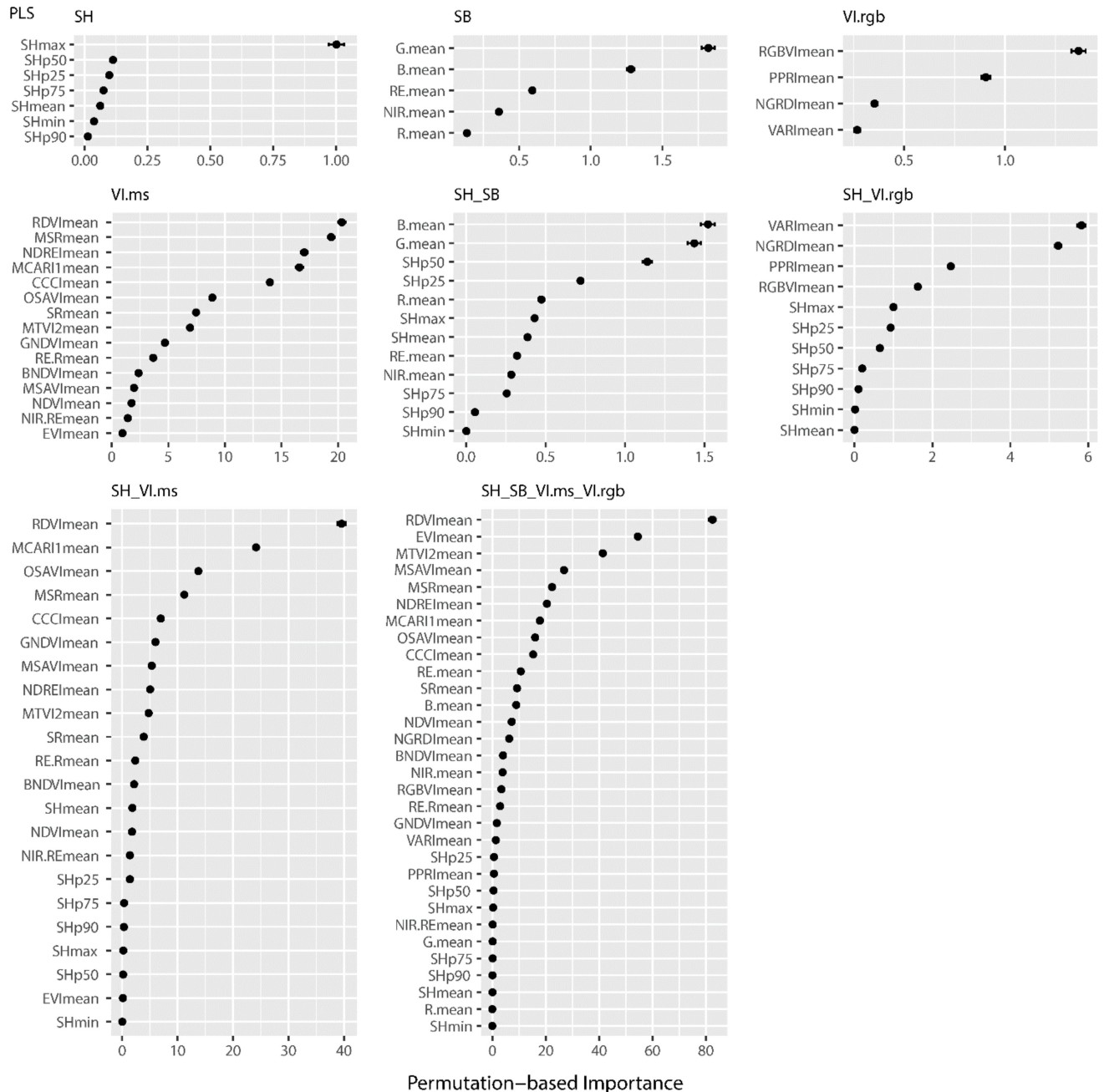

**Figure A4.** VIP of N% estimation with PLS.

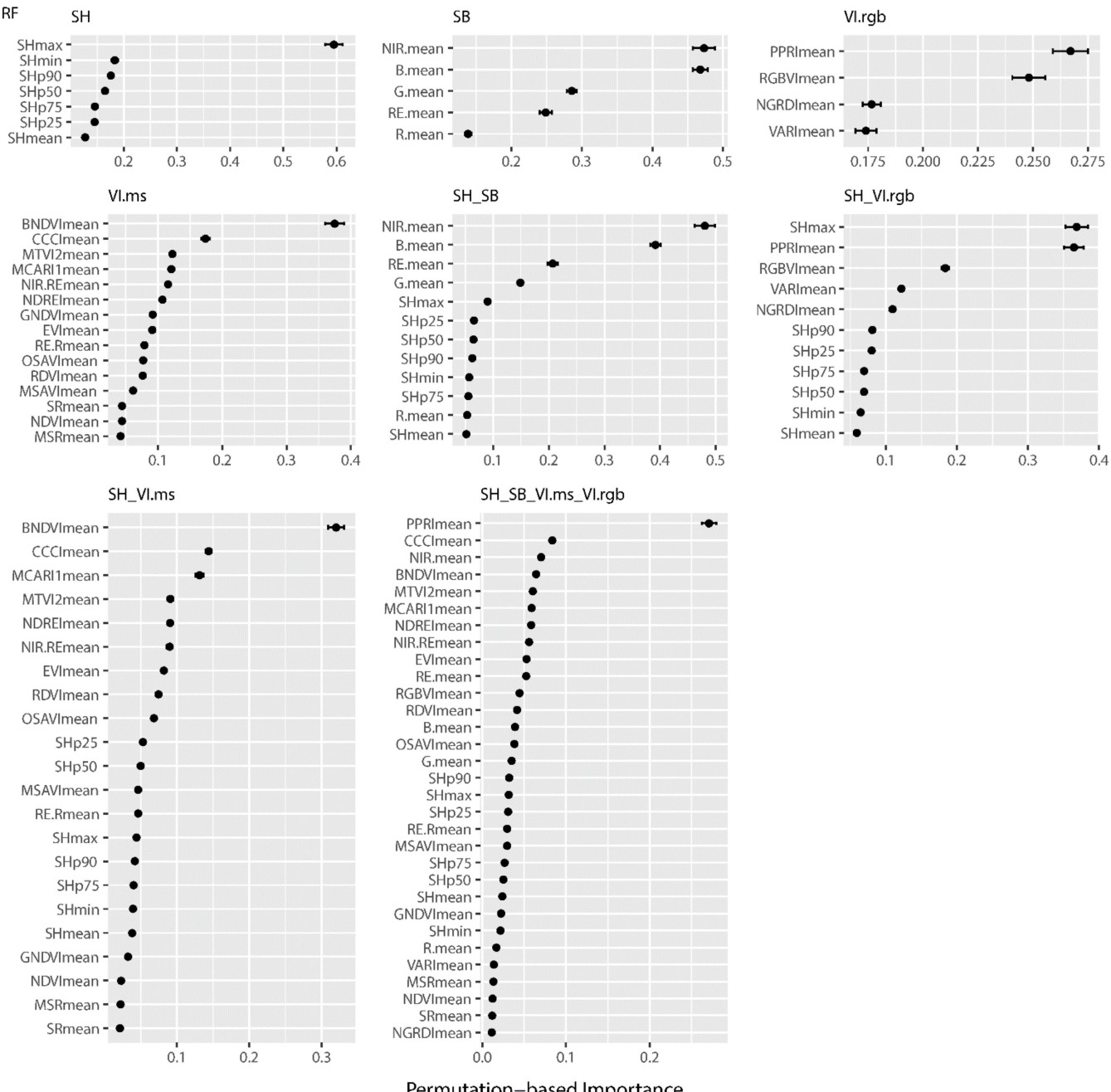

**Figure A5.** VIP of N% estimation with RF.

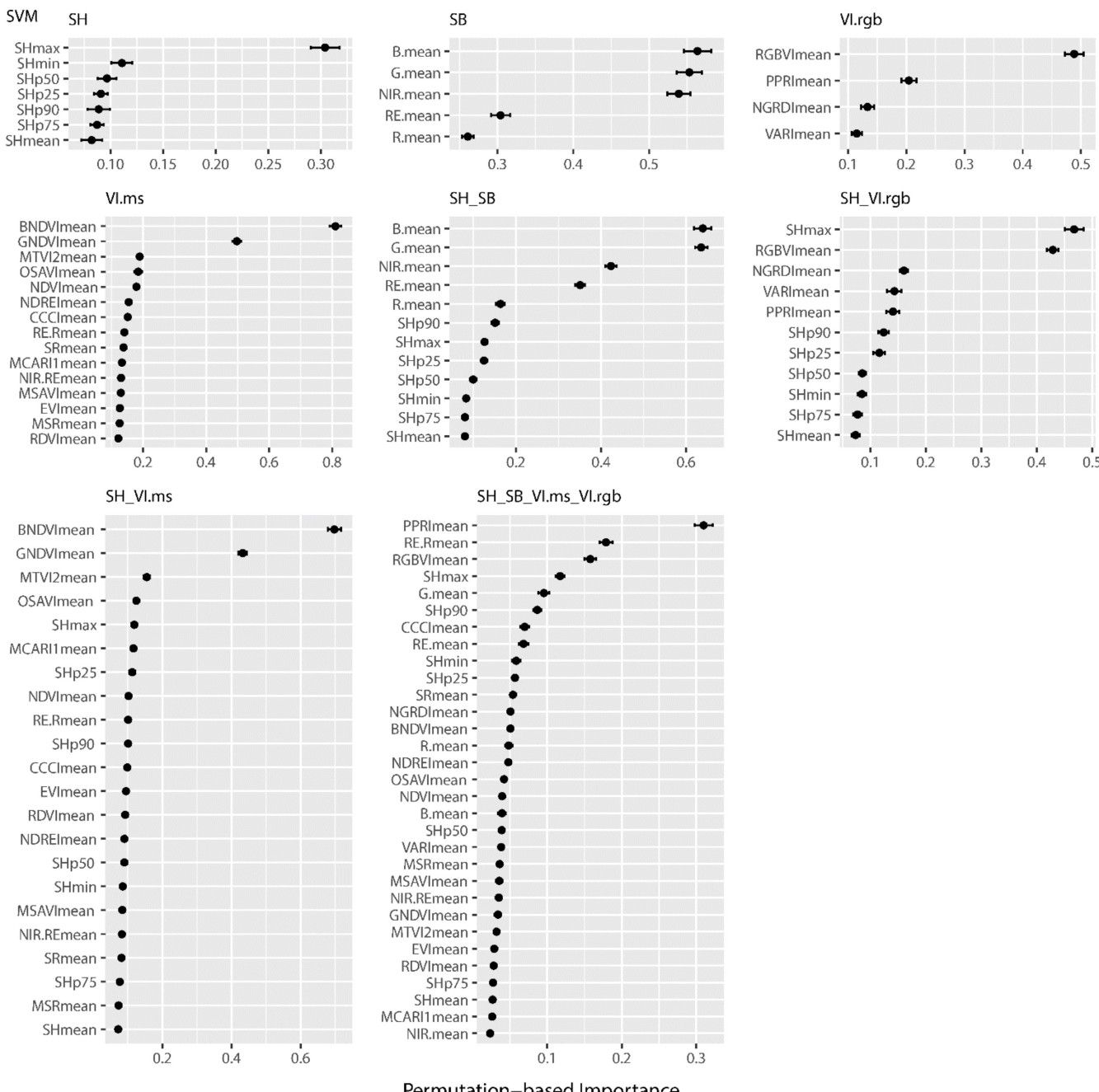

**Figure A6.** VIP of N% estimation with SVM.

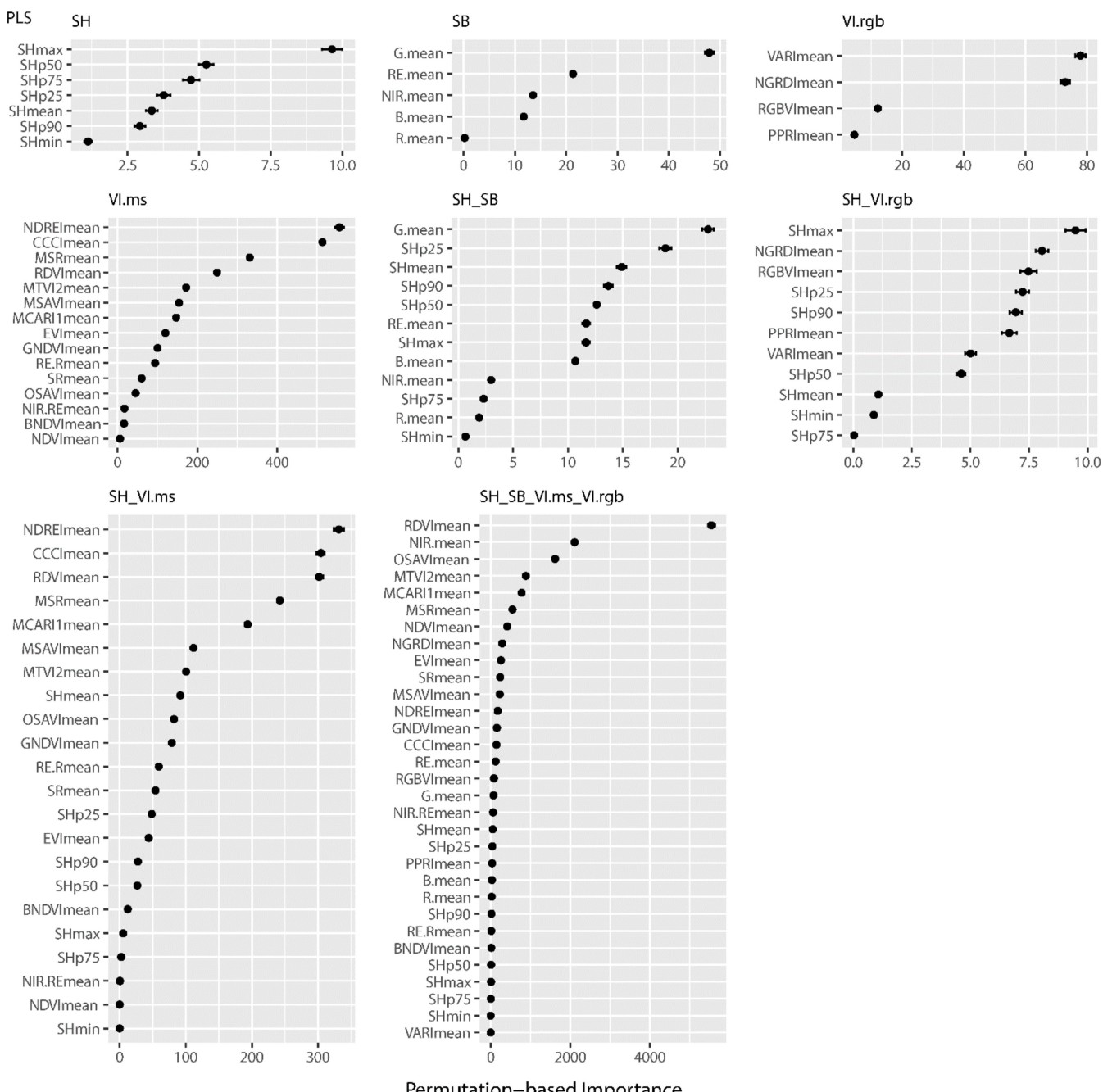

**Figure A7.** VIP of Nup estimation with PLS.

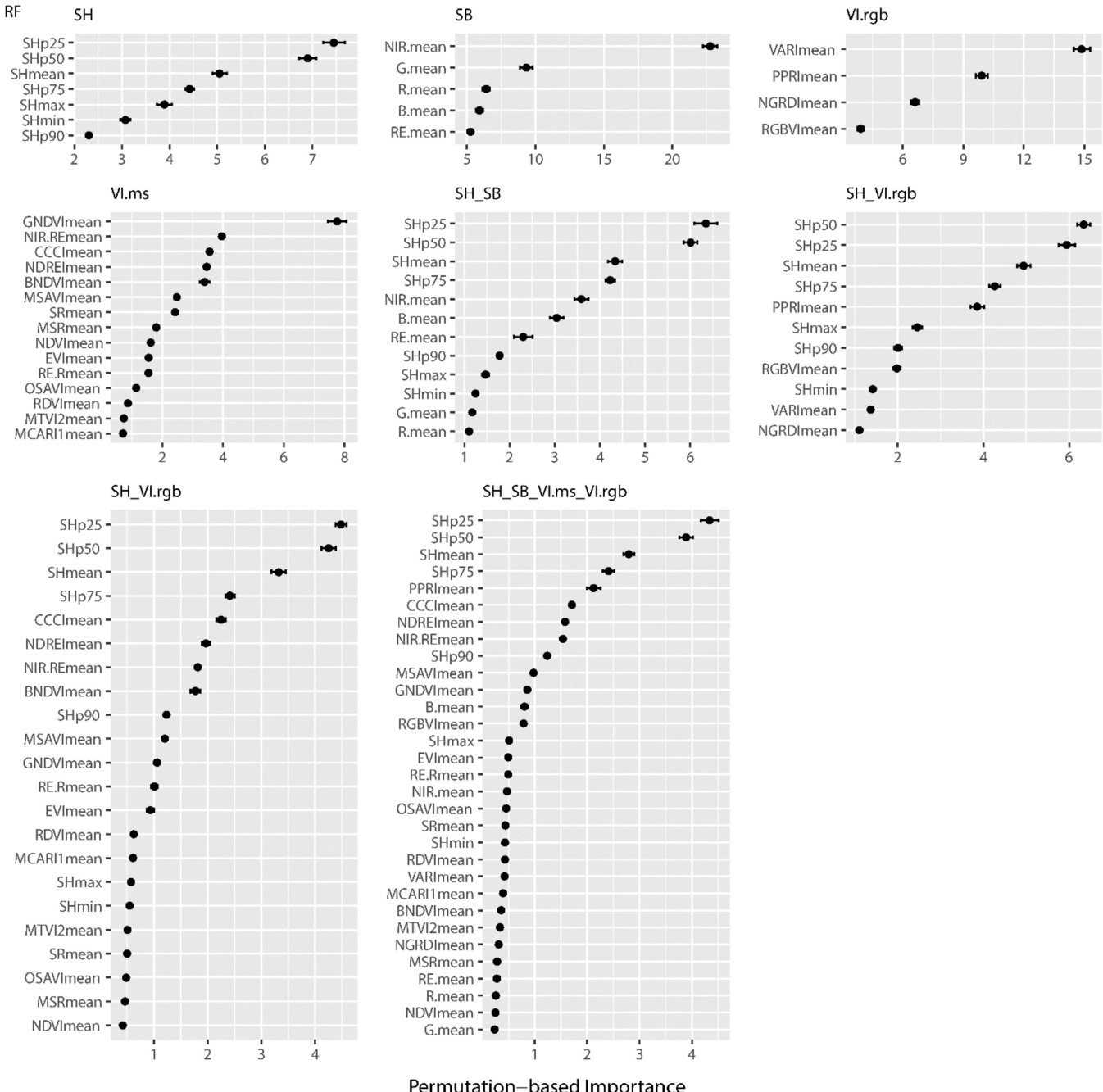

**Figure A8.** VIP of Nup estimation with RF.

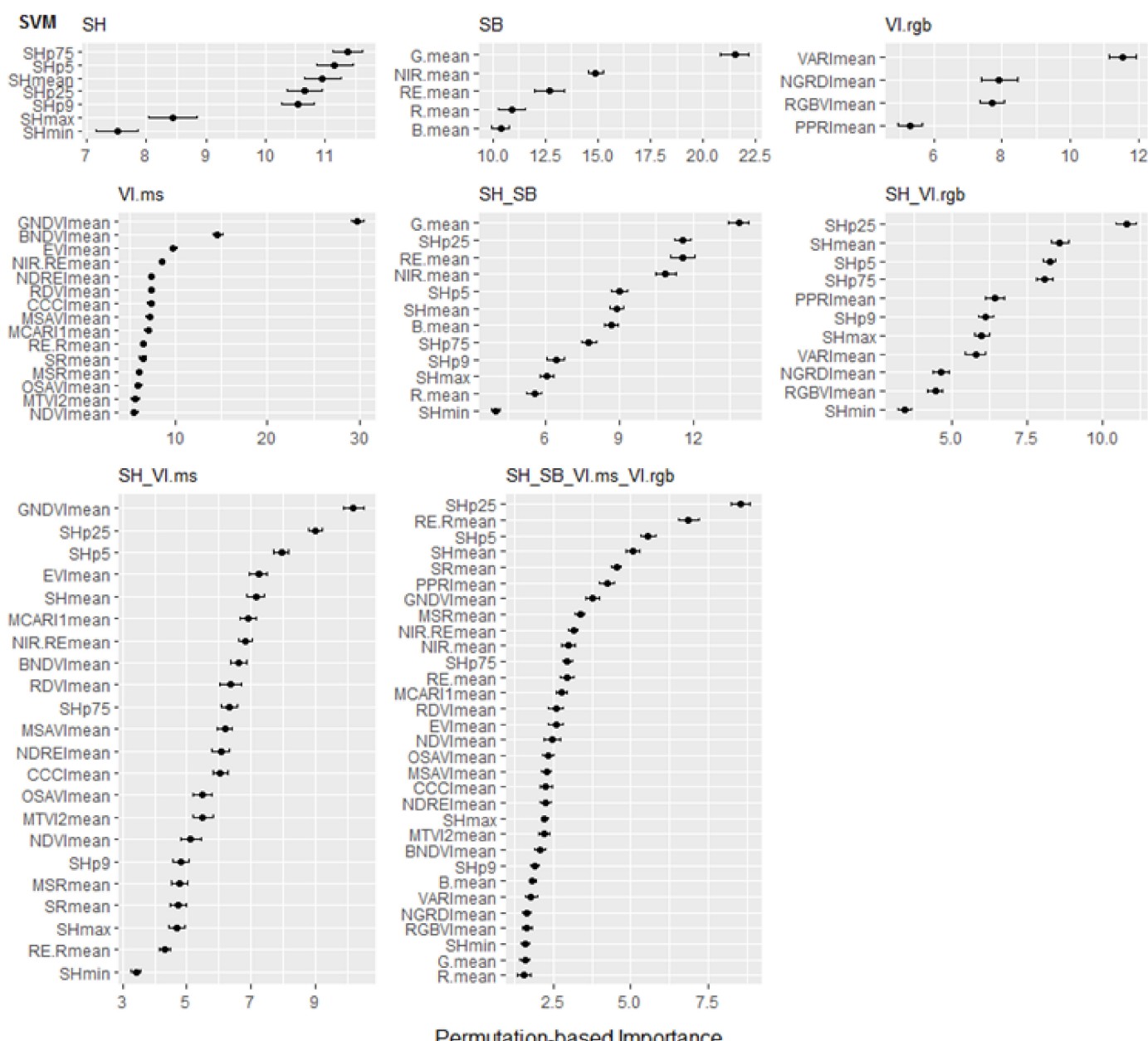

**Figure A9.** VIP of Nup estimation with SVM.

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
