# Peer review of "Herbage Mass, N Concentration, and N Uptake of Temperate Grasslands Can Adequately Be Estimated from UAV-Based Image Data Using Machine Learning"

_remotesensing, doi:10.3390/rs14133066_

Round 1

Reviewer 1 Report

The paper reads well and easy to understand. The introduction, material and methods look good. Minor revision is needed in the writing. Some major changes is required in the conclusion section. I liked the figure, table and charts. It is well presented. 

Reviewer 2 Report

Estimation of multitemporal grassland yield, Nitrogen content and Nitrogen uptake is highly important for farm management practice under economic and environmental aspects The study underlines the potential of a successful usage of sensor fusion with structural and spectral features along with machine learning algorithms. The study was well prepared, and the methods have been clearly developed and presented. The results contribute novel findings which were well discussed and concluded.

I recommend this paper for publication in the journal.

Methods: Could you provide short info on the main species composition at the experimental field?

Did you consider a correction of flight altitude along the slope of the field (using a digital surface or elevation model)?

Figure 2: Why did you use a multicolour palette for the DSM?

If both cameras captured images with 80% side overlap and were dual mounted, do they have the same field of view?

Consider adding a clarifying sentence for what was used in the spectral calibration process: Micasense reflectance panel, sun sensor, NEXTEL panels. I was not totally sure if you included them all.

Why were the base models created at equalization cut? Was there no complete cut of the field right after drone data collection and sample harvest?

Did you also consider deep learning approaches?

Table 4: Row 2 should be stretched.

Table 5: Consider adding (mm-dd) in the ‘date’ column at the row of X, Y, Z

After Figure 13: ‘Additionally, the VI.ms features were among the top ten models.’ What do you mean by top ten models?

After Figure 19: ‘The difference was not significant only for the RF SB model, so the inclusion of the SH metrics yielded no better result for N uptake prediction with the single bands of the MS camera.’ Consider clarifying the sentence, like: ‘The difference was significant for all combinations except the RF SB model […]’

Figure 21: It would make sense to plot the resamples of the reduced feature set along with the full RF model results for better comparison. You could also include R2.

Discussion: ‘Especially the calculation of the base model seems generally to be influenced by the stubble height after the equalization cut, as well as on the remains of the previous harvest, and possible rodent activity.’ As I mentioned before, maybe you could explain why you could not fly for base model creation right after a full harvest of the experimental field. Was there no complete harvest after data/sample collection?

Discussion: ‘[…] however, estimating DMY with CH features only yielded a rRMSE of 30-35%’ Clarify what you mean: CH features only (as pure CH features), or did you mean full model (including ms and CH features) yielded (only) a rRMSE of 30-35%.

Discussion: ‘In comparison, our study yielded significantly lower RMSECV than all of the studies mentioned above. Depending on the algorithm we achieved a median RMSECV around 200 kg ha-1 DMY with a very narrow error range.’ Maybe, the low RMSE could be affected by the DMY range of the samples. As yields in 2018 and 2019 were impacted by drought, the range of DMY could be lower compared to the other studies mentioned.

Discussion: ‘[…] although our study yielded an even lower median rRMSE of 10% for N% and 17% for Nup.’ How was rRMSE calculated (RMSE divided by mean, range, standard deviation or interquartile range)?

Would you expect model performance acceleration by adding texture features?

I hope my suggestions are helpful.

Best Regards.
